# Modeling of Spiral Wound Membranes for Gas Separations—Part II: Data Reconciliation for Online Monitoring

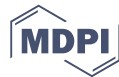

**Diego Queiroz Faria de Menezes** [1,*], **Marília Caroline Cavalcante de Sá** [1],
**Tahyná Barbalho Fontoura** [1], **Thiago Koichi Anzai** [2], **Fabio Cesar Diehl** [2],
**Pedro Henrique Thompson** [2] **and Jose Carlos Pinto** [1]

1   Programa de Engenharia Química/COPPE, Universidade Federal do Rio de Janeiro,
    Rio de Janeiro CEP 21941-972, RJ, Brazil; marilia@peq.coppe.ufrj.br (M.C.C.d.S.);
    tahyna@peq.coppe.ufrj.br (T.B.F.); pinto@peq.coppe.ufrj.br (J.C.P.)
2   Centro de Pesquisas Leopoldo Américo Miguez de Mello—CENPES, Petrobras—Petróleo Brasileiro SA,
    Rio de Janeiro CEP 21941-915, RJ, Brazil; tanzai@petrobras.com.br (T.K.A.);
    fabio.diehl@petrobras.com.br (F.C.D.); pedrothompson@petrobras.com.br (P.H.T.)
*   Correspondence: dmenezes@coppe.ufrj.br; Tel.: +55-21-98807-7489

**Abstract:** The present work presents a methodology based on data reconciliation to monitor membrane separation processes reliably, online and in real time for the first time. The proposed methodology was implemented in accordance with the following steps: data acquisition; data pre-treatment; data characterization; data reconciliation; gross error detection; and critical evaluation of measured data with a soft sensor. The acquisition of data constituted the slowest stage of the monitoring process, as expected in real-time applications. The pre-treatment stage was fundamental to assure the robustness of the code and the initial characterization of collected data was carried out offline. The characterization of the data showed that steady-state modeling of the process would be appropriate, also allowing the implementation of faster numerical procedures for the data reconciliation step. The data reconciliation step performed well, quickly and consistently. Thus, data reconciliation allowed the estimation of unmeasured variables, playing the role of a soft sensor and allowing the future installation of a digital twin. Additionally, monitoring of measurement bias constituted a tool for measurement diagnosis. As shown in the manuscript, the proposed methodology can be successfully implemented online and in real time for monitoring of membrane separation processes, as shown through a real dashboard web application developed for monitoring of an actual industrial site.

**Keywords:** membrane; data reconciliation; real-time; online; monitoring

## 1. Introduction

A common problem in oil production is the excess of $CO_2$ gas present in natural gas streams. The first and most notorious issue is related to the emission of this gas into the environment. However, in addition to the possible environmental problems, the excess of $CO_2$ in oil streams can cause problems in the process plant, such as freezing due to pressure drop in compression and cooling sections of the plant and corrosion of metal pipelines [1]. According to an ANP (Brazilian National Agency of Petroleum, Natural Gas and Biofuels) resolution, commercial natural gas must contain a maximum of 3% (mol) of $CO_2$ [2]. Therefore, a possible solution to deal with the produced $CO_2$ is the reinjection of $CO_2$ into the oil well, which may also allow the increase of the productivity of the well. This can certainly minimize environmental impacts and problems in natural gas process plants.

Therefore, the $CO_2$ separation constitutes a fundamental step during the treatment of natural gas in oil production fields.

Different physical/chemical processes can be used to separate $CO_2$ from natural gas, such as cryogenic distillation, absorption, or membrane reverse osmosis processes [1,3]. Particularly, the removal of $CO_2$ from natural gas with help of membrane separation processes has been used since 1981 [4]. However, applications were initially limited because of intrinsic economic risks associated with the oil production activity and operation constraints related to membrane separations. Nevertheless, the scenario has been changing due to advantages related to the lower energy consumption, low capital investment, low operating costs, and more compact nature of these pieces of equipment [5–7].

Given the increase of the industrial importance of membrane separation processes, demands for development of mathematical modeling, simulation, optimization, control, statistical data treatment, and online monitoring procedures have also increased, as these techniques are fundamental for design and monitoring of chemical processes. As a consequence, the performance of the analyzed process can be evaluated more precisely and monitored, allowing the detection of failures in line and in real time. Based on these technologies, risks and time required for decision-making can be minimized [8].

Based on the previous paragraphs, the main objective of the present work is to develop and implement a web application that makes possible the online and real-time monitoring of membrane $CO_2$ separation processes on an industrial scale for the first time, based on rigorous numerical and statistical procedures. The application can also be used to provide information about unmeasured variables (soft sensor) and to diagnose the occurrence of gross error measurements and instrument malfunctioning. The proposed methodology comprises the following stages: (i) pre-treatment and characterization of process data; (ii) data reconciliation of process data to minimize measurement uncertainties, with the aid of mass balance equations; (iii) detection of systematic deviations for identification of process malfunctions; and (iv) observation of unmeasured variables (soft sensor or digital twin). Finally, the proposed data acquisition and visualization system is implemented online for successful monitoring of an actual industrial membrane separation site in real time for the first time.

*1.1. Data Rectification*

Technological and computer advancement have allowed the wide, easy, and fast access to process data of industrial plants. As a matter of fact, access to actual data are extremely important for real-time monitoring and optimization of production units [9]. The dynamic monitoring of a plant, unit or industrial equipment is increasingly necessary to improve product quality, enhance process safety, and reduce energy costs and waste; however, the acquired information must be reliable and validated with physical process constraints, as the reliability of the data are of paramount importance for any decision-making related to the analyzed process [10]. Nevertheless, process measurements are subject to errors and fluctuations due to intrinsic imprecision, degradation, malfunction, improper installation, poor calibration, and failure of measurement instruments. Additionally, human errors associated with operation and calibration, or gross errors related to the operation of the process, can result in data that do not represent the process reliably. Consequently, measured data are not expected to satisfy physical constraints precisely and are not expected to comply with conservation laws (mass, momentum, and energy balances) [11]. For these reasons, process controllers and data acquisition systems, if not treated properly, can cause the plant to operate at sub-optimal or unsafe operation conditions. In addition, decision-making based on unreliable data can lead to the occurrence of industrial accidents, reduction of product quality, and financial losses [12]. Therefore, the use of data rectification procedures can be essential to improve the quality of the information contained in the data, and consequently provide a margin of reliability for the control and optimization of the process in real time.

Data rectification procedures usually comprise three steps: variable classification; gross error detection (GED); and data reconciliation (DR). Among these three steps, DR and GED are the ones studied most often and applied more frequently in data rectification schemes [13].

### 1.1.1. Data Reconciliation and Gross Error Detection

The variable classification step determines whether the available information is sufficient to solve the DR problem and identify the sets of observable variables (measured and unmeasured variables, which can be estimated using the other measured variables and the process constraints) and unobservable variables (unmeasured variables that cannot be estimated). This way, the variable classification step makes possible data set size reduction in order to include only the relevant variables that can be observed and used to build the mathematical model of the process, reducing the size of the process database [14].

The GED step is performed to identify and/or eliminate/compensate for the occurrence of deviations that do not follow the admitted statistical distribution of errors. Gross errors can be caused by poor calibration of the measuring instruments, deterioration of the sensors, power surges, among other causes described previously. However, in order to obtain accurate estimates of parameters and variables, the negative influence of gross errors must be minimized or eliminated. The use of robust estimators has been frequently suggested to eliminate the negative effect of gross errors, often implemented simultaneously with DR to avoid the use of iterative and computationally intensive numerical procedures [11].

In the DR stage, measured data are adjusted in a statistically coherent manner by an estimator, which frequently is based on the maximum likelihood principle, with the support of a statistical distribution admitted a priori for measurement fluctuations. According to the DR technique, adjusted data must satisfy the conservation laws and other constraints imposed on the system, maximizing the probability of occurrence of that particular measurement and, simultaneously, respecting the mathematical model of the process. Thus, more reliable estimates can be obtained for the variables and parameters of the process [13,15]. Traditionally, the normal distribution is assumed to be valid, which results in the Weighted Least Squares (WLS) estimator [16]. For illustrative purposes, data rectification applications in the industry can be implemented as shown in Figure 1.

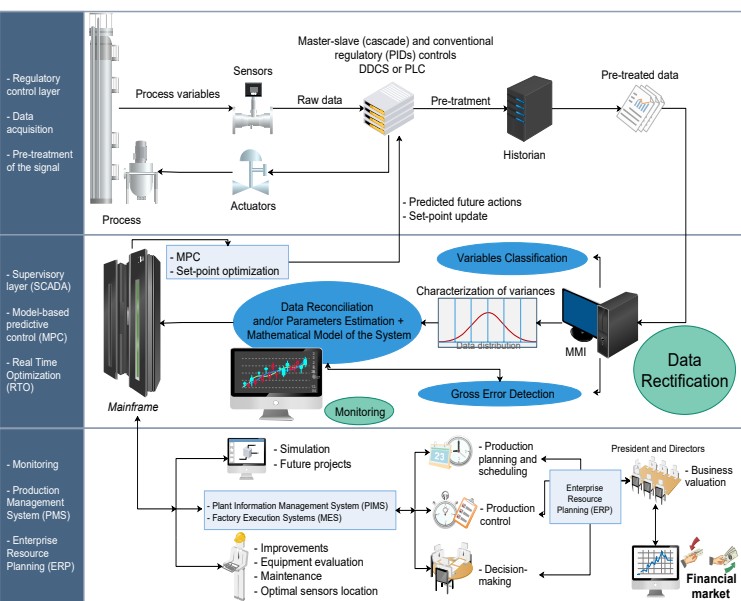

**Figure 1.** Illustrative representation of industrial data rectification applications [12,17].

Kuehn and Davidson [18] were the pioneers in using DR procedures in chemical engineering processes. Since then, many works have proposed the use of DR procedures for monitoring of industrial processes, although the vast majority of the published material investigates simulated processes that operate at steady-state conditions. Investigations of actual dynamic processes in real time and using actual data are scarce and have never been performed for industrial membrane separations [14,19]. Therefore, the present work contributes to the development of DR procedures through the successful implementation of an original application in an actual industrial environment and using real data in real time to perform the proposed analyses. In addition, the present work shows that similar DR applications can be implemented in many industrial membrane separation environments using simple computational resources in real time.

### 1.2. Membrane Separation Process

Membranes constitute excellent alternatives for gas separations due to their low installation and maintenance costs. In the industrial environment, membranes are usually organized in modules with spiral-wound or hollow fiber geometries. Hollow fiber separation units are normally applied to relatively smaller fluxes when compared to spiral-wound modules. On the other hand, spiral-wound modules are cheaper, capable of handling higher operating pressures, and are more resistant to scaling, as particles present in the feed gas stream can block the fine membrane fibers [20,21]. Because of that, the spiral-wound units are largely used in industrial gas separation processes. In a previous work of our group, a mathematical model based on a phenomenological approach for a leaf of a spiral-wound membrane was developed. The model was validated in four case studies of common gas separations, with very good performance and robustness. Furthermore, it allowed the prediction of flow rates and concentrations along the membrane leaf, which are important features for the understanding of membrane operation processes. In addition, a discretization method was proposed to solve the model, which proved to be faster and more efficient than the shooting method [22]. It is also important to emphasize that industrial spiral-wound membrane separation units for $CO_2$ applications are formed by several leaves, which are wounded onto a central perforated collecting tube, forming one modular separation element. These elements can be arranged in series to build a membrane separation tube. Then, tubes can be organized in parallel to form a bank. Finally, the banks can be aligned in parallel to compose a membrane separation train, while the trains can be arranged in parallel to form a stage [21]. Figure 2 shows an example of this kind of unit.

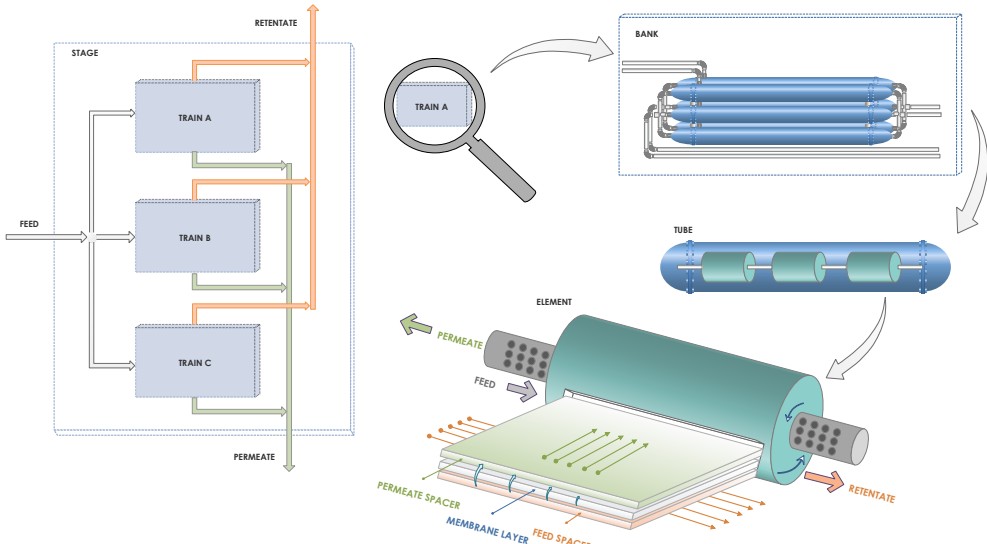

**Figure 2.** Schematic representation of a spiral-wound membrane unit.

1.2.1. Data Reconciliation in the Membrane Separation Process

Currently, very few papers are somehow involved with Data Reconciliation in Membrane Separation Processes. However, the work developed by Lashkari and Kruczek [23] showed that it is possible to reconcile the properties of the membrane separation process, using data related to the process dead time, which is affected by the resistance that the membrane offers to the gas flow. In fact, this work developed procedures to estimate effective permeabilities and diffusion coefficients, allowing the modeling of position-dependent resistance effects, using a lab-scale unit as an example.

Crivellari [19] proposed a model to simulate the separation of $CO_2$ from natural gas through a counter-current spiral wound polymer membrane. The model was used to analyze the influence of distinct variables on process operation conditions and was based on phenomenological balance equations. The model was validated with data collected from the literature and some industrial sites. Moreover, a DR procedure was used to treat the available data and estimate the model parameters; however, the study was implemented offline and did not allow any sort of real-time monitoring of the process operation.

Based on the previous discussions, the present work pioneers the use of DR procedures to monitor membrane separation processes reliably, online and in real time. For this purpose, a membrane plant located in one of the Petrobras Offshore Units was used during this paper. Finally, it is worth noting that, for reasons of industrial confidentiality, numerical results are presented in normalized form.

## 2. Methods

The methodology implemented in the present work comprises six stages: data acquisition; data pre-treatment; data characterization; data reconciliation; gross error detection, and process monitoring (soft sensor or digital twin). The first stages of the implemented procedure involved pre-treatment and characterization of the data. As a matter of fact, proper understanding of some characteristics of the data are fundamental for adequate implementation of the data reconciliation stage [24]. The initial characterization of the data was performed offline and using historical data available in the data acquisition system of the industrial site. The available data were used to determine appropriate sampling periods (based on process response times) and calculate measurement variances (used to formulate the estimation problem) and variable correlations (to characterize independence of measuring devices). Variable classification was also performed to determine the sets of observable and unobservable variables (with the help of the proposed model, as described below) [25].

Using the available data and the model equations, the DR procedure (as described in the following paragraphs) was solved offline to validate the proposed procedure and determine some performance indexes. Particularly, a statistical metrics was used to describe the magnitudes of the deviations between measured and reconciled variables. Then, the model was used offline for calculation of unmeasured variables, providing the soft sensor (or digital twin) response. Finally, the proposed procedures were implemented online and in real time.

The numerical procedures and codes were developed and implemented in Python 3.7.6 (Python Software Foundation, Beaverton, OR, USA) and the details of the proposed methodology are explained in the following sections.

### 2.1. Data Acquisition

Data acquisition was performed through direct access to an industrial database, using standard Plant Information (PI) resources. After performing the numerical operations, a file was saved with the measured, reconciled, estimated, and calculated variables. Storage was performed during monitoring, to avoid accumulation of data in the computer memory and save the relevant information in real time.

The "pandaspi" library was utilized to provide communication between Python and PI, transferring the information directly to a data frame [26,27]. By using these resources, the data acquisition process became very simple and practical, as access to the data depended only on the login,

password, tags of the desired variables, the size of the sample window, and the sampling frequency. The time interval selected for offline analyses was equivalent to two weeks with a sampling frequency of 5 min, which provided a sufficiently high number of points for the execution of the pre-treatment step. An additional number of data points did not provide any significant improvement of the preliminary analyses in the considered case so that this should not be regarded as a drawback of the proposed analysis.

*2.2. Data Pre-Treatment*

Data pre-treatment comprised the following steps: reading the Excel Workbookfiles in UTF-8 encoding; pre-treatment of raw data spreadsheets; and data storage in Hierarchy Data Format version 5 (HDF5).

During real-time monitoring, the data reading step was performed as described in the previous section. Data storage in HDF5 format was also performed as described previously. When the file is saved in HDF5 format, reading presents better performance, since reading data directly from the Excel spreadsheet can be too slow for real-time applications [28,29].

The pre-treatment stage organized the raw data from the PI into data frames (Pandas library) [26,27]. The treatment followed the following steps: standardize the indices (day/hour/minute of the samples); variables that contain some string must be replaced (such as on/off by 0/1 and error notices by NaN, not a number); chronologically sort the data and replace missing data (NaN) with neighbors (back and forward fill). This last step is crucial to assure that the acquired data window does not contain missing data, which can make the calculation of variances difficult in the acquisition window. Unit conversions, normalization of concentrations and calculation of standard deviations and variances completed this step.

Data storage must be carried out after data reconciliation and energy balance calculations. The file was saved with the measured, reconciled, estimated, and calculated variables. As already explained, storage was performed during monitoring, to avoid accumulation of data in the computer memory and save the relevant information in real time.

*2.3. Data Characterization*

During this step, the procedures that must precede the data reconciliation task, such as the visualization and statistical characterization of the data, must be carried out. The characterization of the data was performed in accordance with the following steps:

- Visualization of variables;
- Selection of variables of interest;
- Quantification and visualization of missing data (NaN);
- Construction of the boxplots of the variables of interest [30];
- Analysis of the variance spectra [31];
- Calculation of variances and correlations.

The main pursued objectives during this stage were the proper characterization of the data quality, the analysis of the operation dynamics, and the characterization of the stationarity of the phenomenological process model. The analyses of missing data and boxplot properties can indicate the quality of collected data during the selected time period and the number of gross errors in the data base.

The boxplot is a graph used to assess the empirical distribution of data. The boxplot is a non-parametric analytical technique, which shows the measurement variations within a statistical population without making any assumption about the underlying statistical distribution. The box is usually built with the first ($Q_1$) and third ($Q_3$) quartiles (50% of the data) and the median ($Q_2$). The lower and upper lines extend, respectively, from the lower quartile to the lowest value not lower

than the lower limit (*LL*), and from the upper quartile to the highest value not higher than the upper limit (*UL*). The limits can be calculated according to Equations (1) and (2):

$$LL = \max \left[ \min(data), \quad Q_1 - 1.5(Q_3 - Q_1) \right] \tag{1}$$

$$UL = \min \left[ \max(data), \quad Q_1 - 1.5(Q_3 - Q_1) \right] \tag{2}$$

The value 1.5 is tuned to capture 99.7% of the data between the lower and upper limits, assuming the normal distribution [30]. In summary, the boxplot identifies the regions in the variable domain where 50% and 99.7% of the data are located. The points outside these limits are tagged as outliers. Boxplots were plotted using the Seaborn library available for Python [32].

The analysis of the variance spectra shows how the process variance depends on the size of the sampling window. This type of spectrum provides information about the various sources that contribute to the signal of a variable, including noise/measurement errors (short window sizes) and intrinsic process variations (large window sizes). The variance spectrum can be defined as a set of variances calculated while some variable related to them evolves [31]. The spectrum of variances for short sampling windows are controlled by the variances of the measuring instrument. This way, the best estimate for the variance of the measuring device can be calculated using the variance spectrum with short sampling windows. However, the use of very short sampling windows may not reveal the actual variability of the data, due to poor measurement quality. The spectrum for sufficiently large sampling windows captures the variability of the entire process, including operational changes. More details about the usefulness of this technique for characterization of process data are provided by Feital and Pinto [31].

Analyzing the correlations, autocorrelations, and cross-correlations between pairs of variables of interest allows the characterization of stationarity, seasonality, and regions of operation of the process [33,34]. Observing correlations between input and output flows can indicate process stationarity. The absence of correlation can indicate the occurrence of significant nonlinearity or dynamics in the process response [35]. For dynamic responses, cross-correlation can capture the lags between the actions on the input variables and the steady-state of the output variables. Obviously, the identification of dynamic correlations among the many variables of the system may indicate the necessity to build and implement dynamic models for more accurate representation of the available data. For this reason, proper characterization of stationarity can be important for more successful implementations of monitoring procedures [36].

### 2.4. Data Reconciliation

The DR procedure consists of solving an optimization problem characterized by an Objective Function (OF) that must be minimized while respecting certain restrictions (model). The OF of the DR is often proposed as the maximum likelihood estimator resulting from a statistical distribution of measurement errors, which is commonly adopted as the normal distribution. After application of the principle of maximum likelihood, the normal distribution results in the WLS estimator. This way, the problem originally formulated by Kuehn and Davidson [18] can be written in accordance with Equations (3)–(7) [17]:

$$\hat{\underline{z}} = \min_{\hat{\underline{z}}} \frac{1}{2} \left[ \underline{z} - \hat{\underline{z}} \right]^T \underline{\underline{V}}^{-1} \left[ \underline{z} - \hat{\underline{z}} \right] \tag{3}$$

subject to:

$$\underline{h}(\hat{\underline{z}}, \underline{u}) = \underline{0} \tag{4}$$

$$\underline{g}(\hat{\underline{z}}, \underline{u}) \geq \underline{0} \tag{5}$$

$$\hat{\underline{z}}^L \leq \hat{\underline{z}} \leq \hat{\underline{z}}^U \tag{6}$$

$$\underline{u}^L \leq \underline{u} \leq \underline{u}^U \tag{7}$$

where $\hat{\underline{z}}$ is the vector of the reconciled variables; $\underline{z}$ is the vector of the measured variables; $\underline{V}$ is the matrix of variances for measurement errors; $\underline{u}$ is the vector of the unmeasured variables (observable); $\underline{h}()$ is the vector of linear or nonlinear algebraic constraint equations; $\underline{g}()$ is the vector of the inequalities of linear or nonlinear algebraic restrictions; $\hat{\underline{z}}^L$ and $\hat{\underline{z}}^U$ are the upper and lower parameter vectors of the $\hat{\underline{z}}$ vector and $\underline{u}^L$ and $\underline{u}^U$ are the upper and lower parameter vectors of the $\underline{u}$ vector.

For DR problems where the model constraints are linear and all variables are measured, the analytical resolution of the problem can be obtained with help of Lagrange Multipliers [37]. However, the set of individual mass balance equations generate a nonlinear system of equations that involve unmeasured variables, as described below. For this reason, a successive linearization procedure was used to solve this problem [38].

The observability analysis of the system can also be performed during the successive linearization procedure, using QR factorization. The procedure consists of describing the nonlinear model in terms of two matrices: a matrix related to the measured variables and a second matrix related to the unmeasured variables. Therefore, the system is observable if the rank of the matrix of unmeasured variables is equal to the number of unmeasured variables [39].

The following points were considered during the implementation of the proposed model in the industrial site, illustrated in Figure 3:

- Period for preliminary characterization of database: two weeks with sample frequency of 5 min;
- Measured variables (z):

  ○ Flowrates: Total Feed ($F$), Total Retentate ($R$), Train Retentate $A, B$ and $C$ ($R_A, R_B$ and $R_C$) [$kNm^3/h$];
  ○ Components: $C_1$(methane), $C_2$(ethane), $C_3$(propane), $C_6$(hexane), $C_7$(heptane), $C_8$(octane), $CO_2$(carbon dioxide), $iC_4$($i$-butane), $iC_5$($i$-pentane), $N_2$(nitrogen), $nC_4$($n$-butane) and $nC_5$($n$-pentane) in the 3 streams.

- Unmeasured variables (u):

  ○ Flowrate: Total permeate ($P$) [$kNm^3/h$].

*Model* (Component Mass Balance—Steady-State):

$$\underline{h}(\hat{\underline{z}}, \underline{u}) = \begin{cases} \underline{0} = F\underline{y}_F - R\underline{y}_R - P\underline{y}_P \\ 0 = R - R_A - R_B - R_C \\ 0 = 1 - \sum_{i=1}^{nc} \underline{y}_{F,i} \\ 0 = 1 - \sum_{i=1}^{nc} \underline{y}_{R,i} \\ 0 = 1 - \sum_{i=1}^{nc} \underline{y}_{P,i} \end{cases} \tag{8}$$

- Number of points in the study phase (nt) = 3457
- Number of components (nc) = 12
- Number of measured variables at each sampling point (Nm) = $3nc + 5 = 41$
- Number of unmeasured variables at each sampling point (Nu) = 1
- Number of total equations at each sampling point (Nv): $Nm + Nu = 42$

- Number of constraint equations (Nce) = $nc + 4 = 16$
- Number of optimization variables at each sampling point (Nopt): $Nv - Nce = 26$
- Degrees of freedom at each sampling point (DF): $Nm - Nopt = Nm - (Nm + Nu - Nce) = 15$

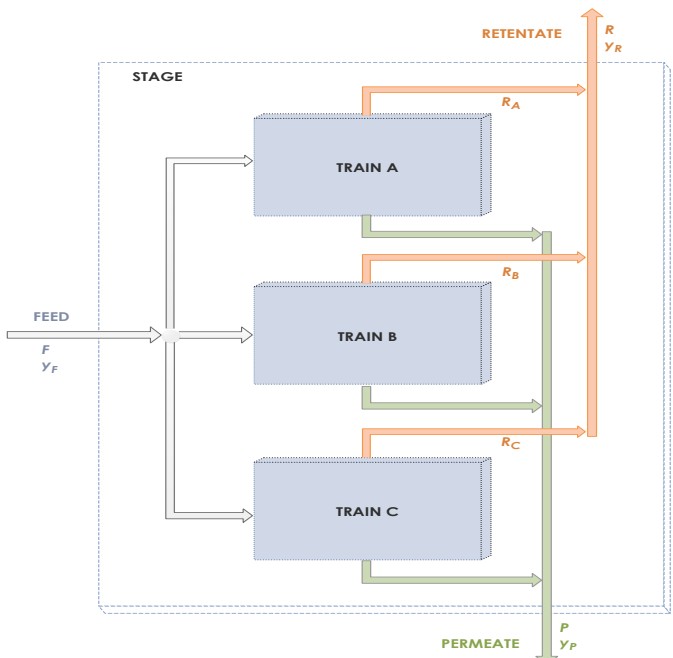

**Figure 3.** Individual mass balances in the envelope around the process.

*2.5. Gross Error Detection*

Gross errors are those originating from non-random events, having little or no connection with the measured value. They may be related to measurements (such as malfunctioning of instruments) or process (such as leaks). Consequently, gross errors invalidate the classic statistical basis of traditional DR methods and undermine the systematic analysis of the data, demanding the implementation of Gross Error Detection procedures for compensation or elimination of gross errors, which may precede the DR step. Gross errors can be further classified as **bias**, **outliers**, and **drifts** [40].

The metrics applied in this step to identify the occurrence of gross errors is related to the statistical test of standard hypotheses for GED [12]. This test is widely used and simple, and also forms the basis for all other classic GED procedures. The test is applied to the data set to determine whether the measurements follow a symmetrical distribution of the measurement errors.

Reilly and Carpani [41] were the first to study the detection of gross errors in process engineering. The authors proposed the use of a statistical test of the type $\chi^2$ (chi-square) based on the residues of the process model, which was called the Global Test (GT).

In the present work, GED was based on the GT, using the sampling window to observe samples and identify the occurrence of gross measurement errors. According to this procedure, the data in the moving window are used to compare the errors of the Weighted Least Squares estimator with the deviations from median values within the window and using the $\chi^2$ function to characterize significant deviations between these computed variances [42].

The outlier effect during DR is removed through compensation. After identification of a possible outlier, the variable value was manipulated to adjust the expected variance. Thus, a moving variance window was implemented to monitor changes in operation, measurement errors, and failures. The statistical test was performed in sampling windows containing at least 20 samples. With this, the median and the standard deviation were calculated and compared with the value of the variable at the current point. Therefore, if the value was more than seven standard deviations apart from

the median, the measurement would be regarded as an outlier and the variance adjustment would be performed.

To monitor the possible occurrence of bias, a dynamic bar graph illustrating the magnitude of errors for each variable was implemented, using the following metrics:

$$Bias_{i,t} = \frac{med(|z_{i,t} - \hat{z}_{i,t}|)}{dp_{i,t}} \tag{9}$$

$$dp_{i,t} = NMAD(|z_{i,t} - \hat{z}_{i,t}|) \tag{10}$$

where $NMAD$ is Normalized Median Absolute Deviation.

### 2.6. Monitoring

Instrumenting the whole process can be very costly and may lead to acquisition of unnecessary and obsolete information. In addition, in some cases, it can be impossible to measure the desired variable. For this reason, some information that is essential for process monitoring can be assessed through models. Therefore, a soft sensor can estimate variables with a mathematical model, in real time, using available plant data, as measured by existing instrumentation. Particularly, the use of plant variables can provide opportunities to improve the performance of a plant [43].

In the analyzed process, the permeate flow was not measured. It must be emphasized that this is not unusual at real industrial sites. However, through DR, this information can be obtained with the help of the model, after the application of the DR procedure. The complete set of temperature and pressure was not available either, which is not unusual at the plant site. In the analyzed case, the pressure of the feed stream and the temperature of the permeate stream were not measured. Therefore, the full implementation of the energy balance in the proposed DR procedure was not viable due to the lack of observability. For this reason, the pressure of the feed stream was evaluated through calculation of the pressure loss in the separation stage, based on design data and the pressure data of the retained stream. In this case, after characterization of the pressure loss, the energy balance equation can be used to estimate the permeate stream temperature, with the aid of Equation (11), calculating the enthalpies of the process streams with the Peng–Robinson equation of state [44]:

$$0 = F.H(T_F, P_F, \underline{y}_F) - R.H(T_R, P_R, \underline{y}_R) - P.H(T_P, P_P, \underline{y}_P) \tag{11}$$

It must be noted that the execution of the numerical procedure in this analyzed case was extremely fast (order of milliseconds), so that the computer hardware exerted little influence on the application. The slowest step of the numerical procedure was data acquisition (and the bottleneck was data transfer and connection speed), so that instrumentation hardware and data handling software constituted the most sensitive parts for this particular real-time application. Despite that, it took only few seconds for data downloading to be complete; consequently, the online and real-time implementation could be performed in a standard notebook equipped with an Intel Core i7 8th gen processor (Intel Corporation, Santa Clara, CA, USA).

## 3. Results and Discussion

### 3.1. Data Characterization

The Missingno library was used to analyze the missing data [45]. In Figure 4, one can observe the data density, visualizing completely the pattern of missing data in the whole set, with the columns representing the variables and lines representing data points of the time series. In addition, a frequency bar that indicates the number of variables measured at each particular timeline can be seen on the right side of the graph.

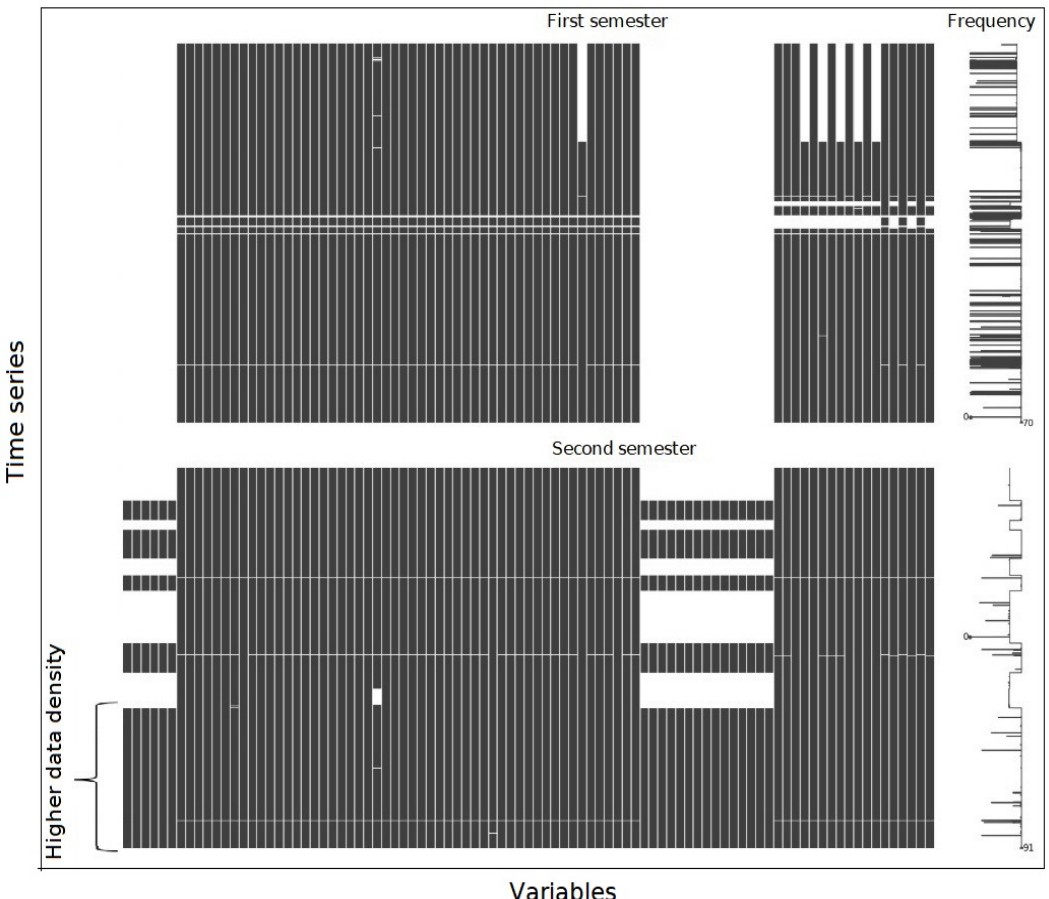

**Figure 4.** Missing data analysis.

The analysis of Figure 4 illustrates how one can select the best data period for the execution of the initial data processing stage. The period with the highest data density allows the analysis of the plant operation with better reliability, avoiding blind periods of instrumentation or shutdown of operation. Therefore, the selection of the period is fundamental for the data characterization stage.

Figure 5 illustrates the boxplot of chromatographic analyses associated with the four main components of the feed stream. This analysis allows the preliminary evaluation of the precision of the instrumentation and/or the variability of the operation. A process with distinct operating points generates multimodal distributions, which requires more involving analysis to qualify the precision of the measurement, such as violin plot (a combination of boxplot and kernel density estimate) [46]. However, during stationary operation periods, the boxplot analysis showed that chromatographic measurements presented good precision, which is also illustrated by the small number of outliers with respect to the total number of 3673 samples.

Figure 6 shows that the "gross errors" followed a downward trend, with highest concentration below the modal value. This is because a drop in the process flowrate was observed during this period. Therefore, in this case, the analysis interprets that the operating changes are "gross error", when they are not. On the other hand, the good behavior of the data indicates that the flowrate was measured with good precision. Instrumentation accuracy was also analyzed after DR, with help of the bias analysis.

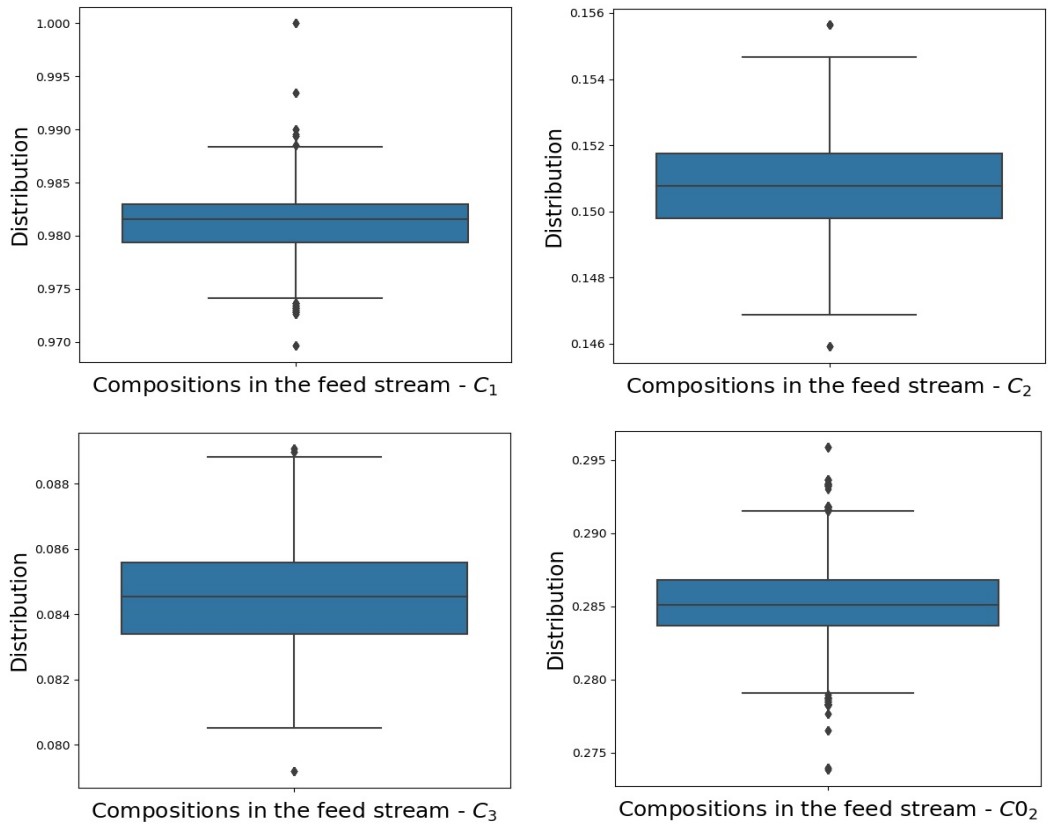

**Figure 5.** Analysis of boxplots for compositions $C_1$, $C_2$, $C_3$, and $CO_2$ in the feed stream.

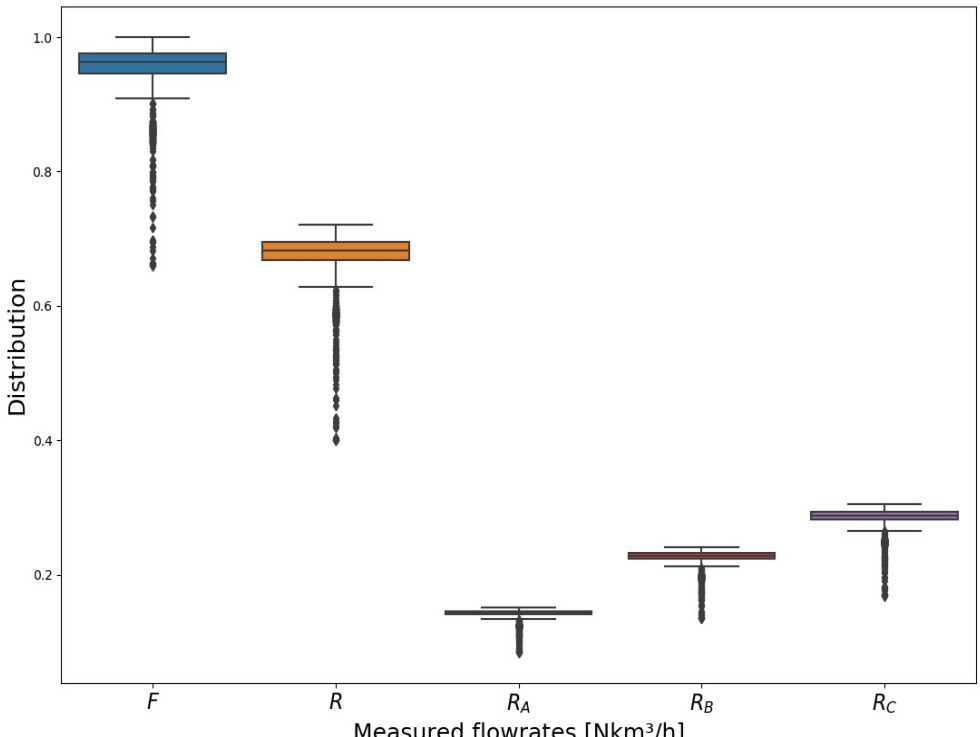

**Figure 6.** Boxplot analysis of flowrates.

Figure 7 illustrates the time series for the compositions of $C_1$, $C_2$, $C_3$, and $CO_2$ in the feed stream, while Figure 8 shows the feed and retentate flowrates, for the same time ranges analyzed in the boxplot.

Therefore, it becomes evident in the case of flowrates that the supposed gross errors actually indicated a change in operation and not a failure of the sensor. In the case of composition, outliers can possibly be assigned to gross errors, although only the DR can allow the proposition of reliable statements about the alleged gross errors.

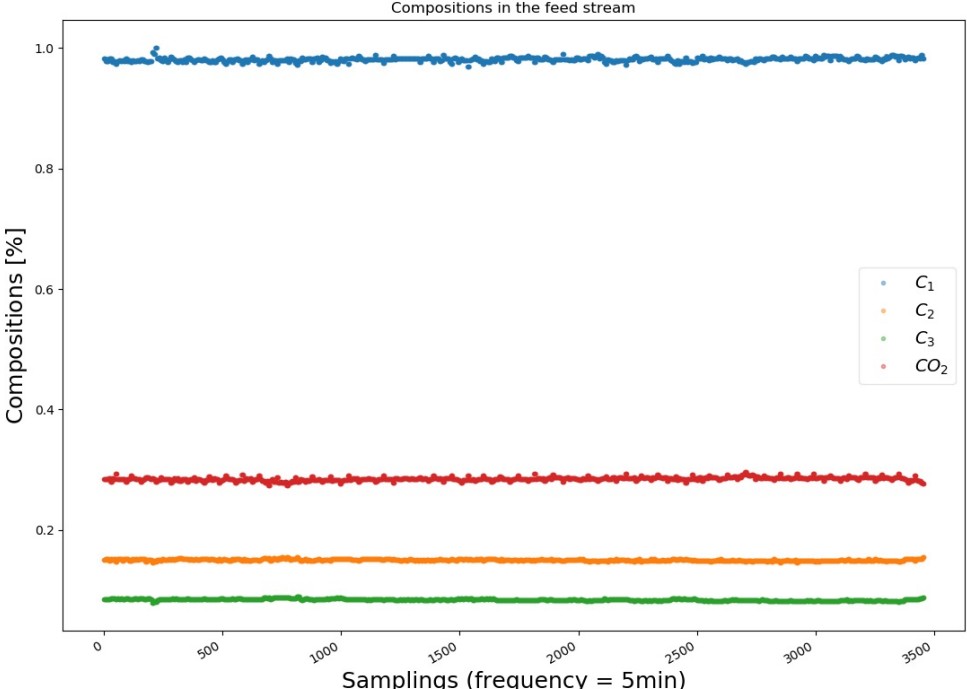

**Figure 7.** $C_1$, $C_2$, $C_3$, and $CO_2$ compositions in the feed stream.

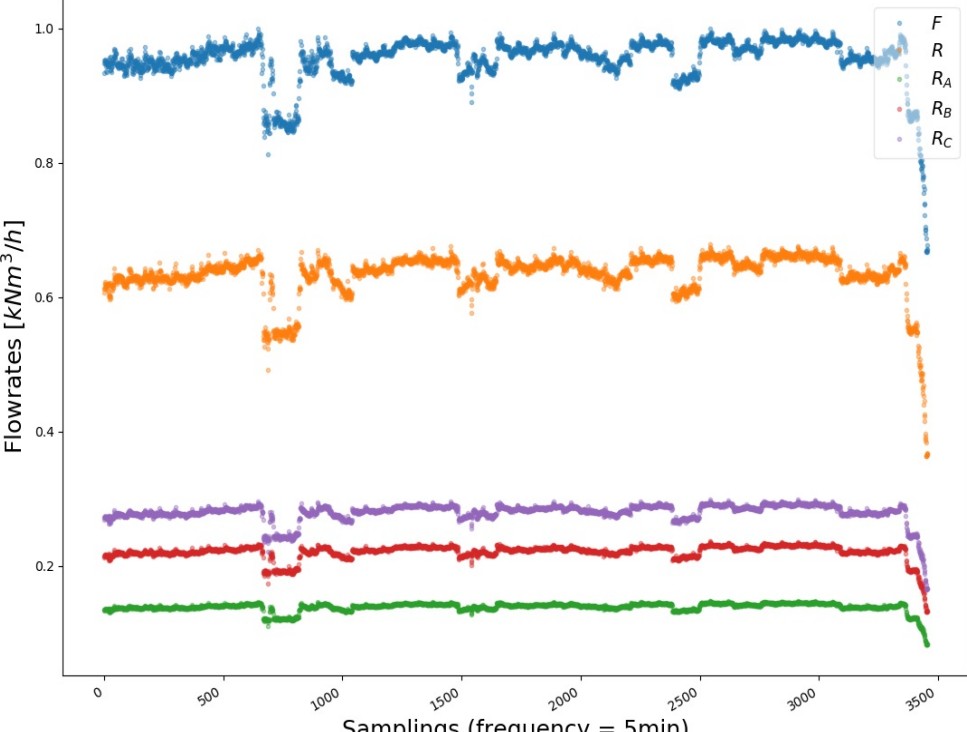

**Figure 8.** Measured flowrates.

An important analysis is related to the observation of the correlations between pairs of variables. Correlations can indicate absence or presence of process stationarity. Figure 9 shows a strong linear correlation between feed, retentate, and permeate flowrates. Strong linear correlation between inlet and outlet flowrates can be an indication of process stationarity [47].

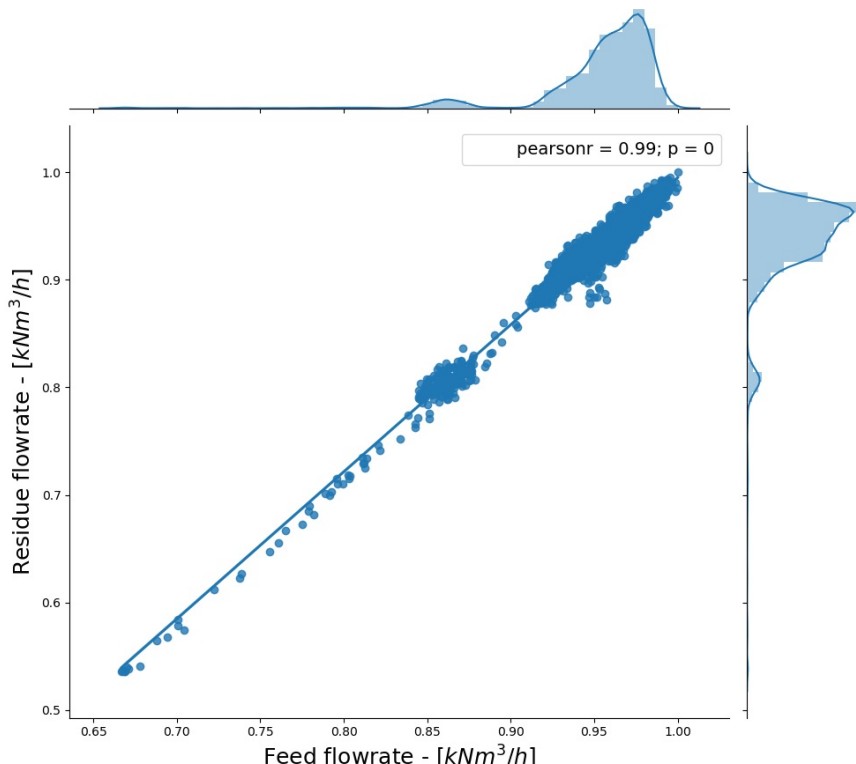

**Figure 9.** Correlation analysis for feed and retentate flowrates.

Figure 10 illustrates the Autocorrelation Function (ACF) and Partial Autocorrelation Functions (PACF) for feed and residue streams. This analysis provides the diagnosis of temporal dependence between the lags of individual variables, which in this case were evaluated for lags ranging from 0 to 50 lags. As shown in Figure 10, the ACF decayed continuously and just one lag caused the appearance of strong correlation (close to 1) in the PACF. Therefore, the process presents very short dynamic memory, indicating the quasi steady-state behavior and constituting an auto-regressive process of order 1 [33].

Figure 11 illustrates the Cross-Correlations Function (CCF) between feed and retentate flowrates up to 50 lag. Cross-correlations decayed slowly for different pairs of variables, indicating that the process operated at quasi steady-state conditions and that responses were much faster than the characteristic sampling times. Therefore, the analyzed membrane separation process could be considered to operate at steady-state. This validated the use of the steady-state mass and energy balance equations in the DR problem. Given the small volumes of most membrane separation modules and the large flowrates of typical industrial plants, this conclusion can probably be extended to other industrial sites, allowing the more general use of the analysed procedures in other industrial facilities.

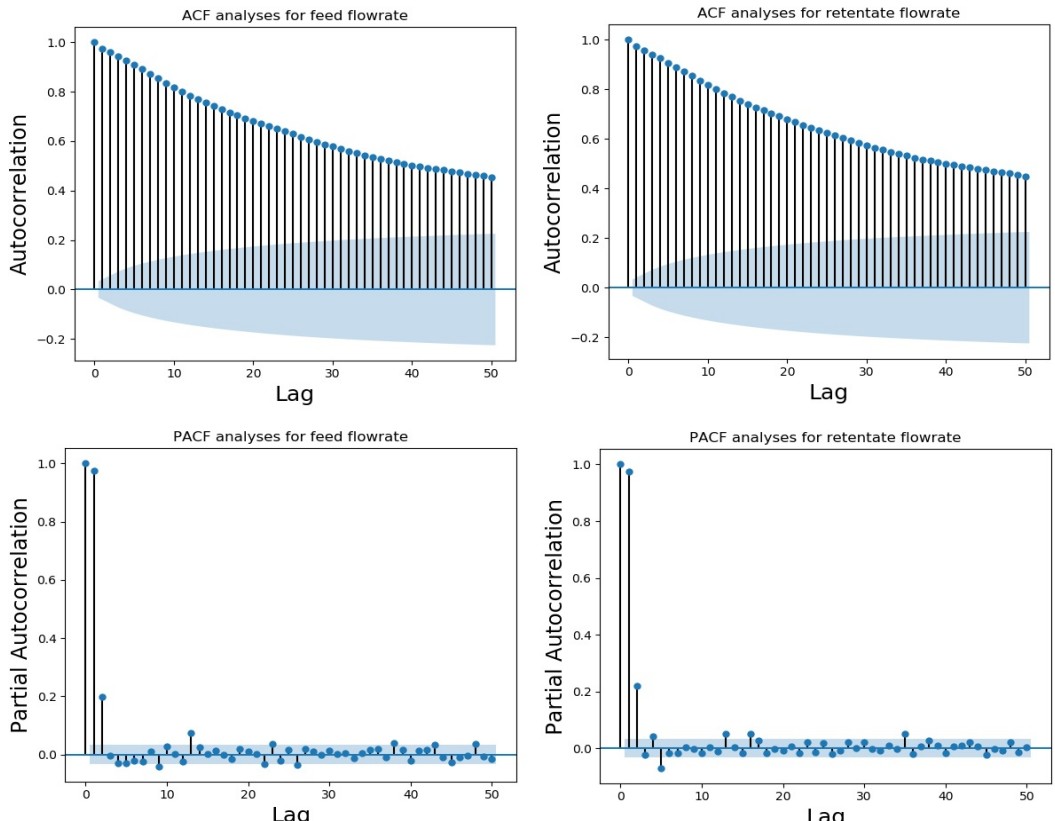

**Figure 10.** ACF (Autocorrelation Function) and PACF analyses (Partial Autocorrelation Functions) for feed and retentate flowrates.

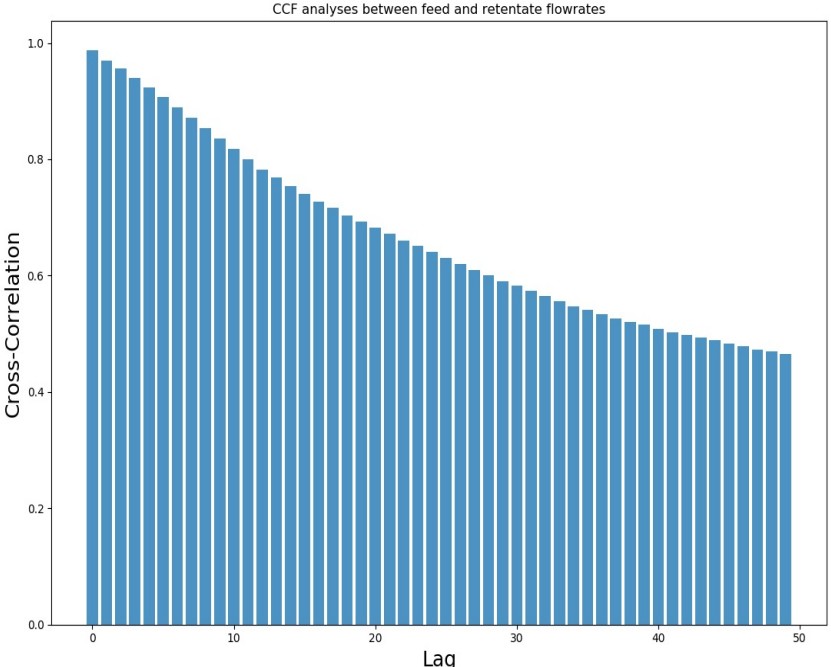

**Figure 11.** CCF analyses between feed and retentate flowrates.

### 3.2. Data Reconciliation

Before starting the DR procedure, the system was analyzed with the help of Variable Classification techniques and the system was classified as observable, indicating that the measured variables can be reconciled and that unmeasured variables (permeate flowrate) can be estimated.

The first DR results were obtained through offline simulations, using a sampling period of two weeks with a sampling interval of 5 min. The data reconciliation performed very well and the problem was solved at average computational speed of 1.7 ms/sample. This result clearly showed that the application could be implemented online and in real time due to the sampling interval of 5 min.

Figure 12 illustrates, as an example, the measured and reconciled data for chromatographic measurements of the four main components in the feed stream. One of the advantages of DR is to restore the resolution of the amplitude signal of the measured value, which can be seen in Figure 12, especially for samplings of the $C_2$ component.

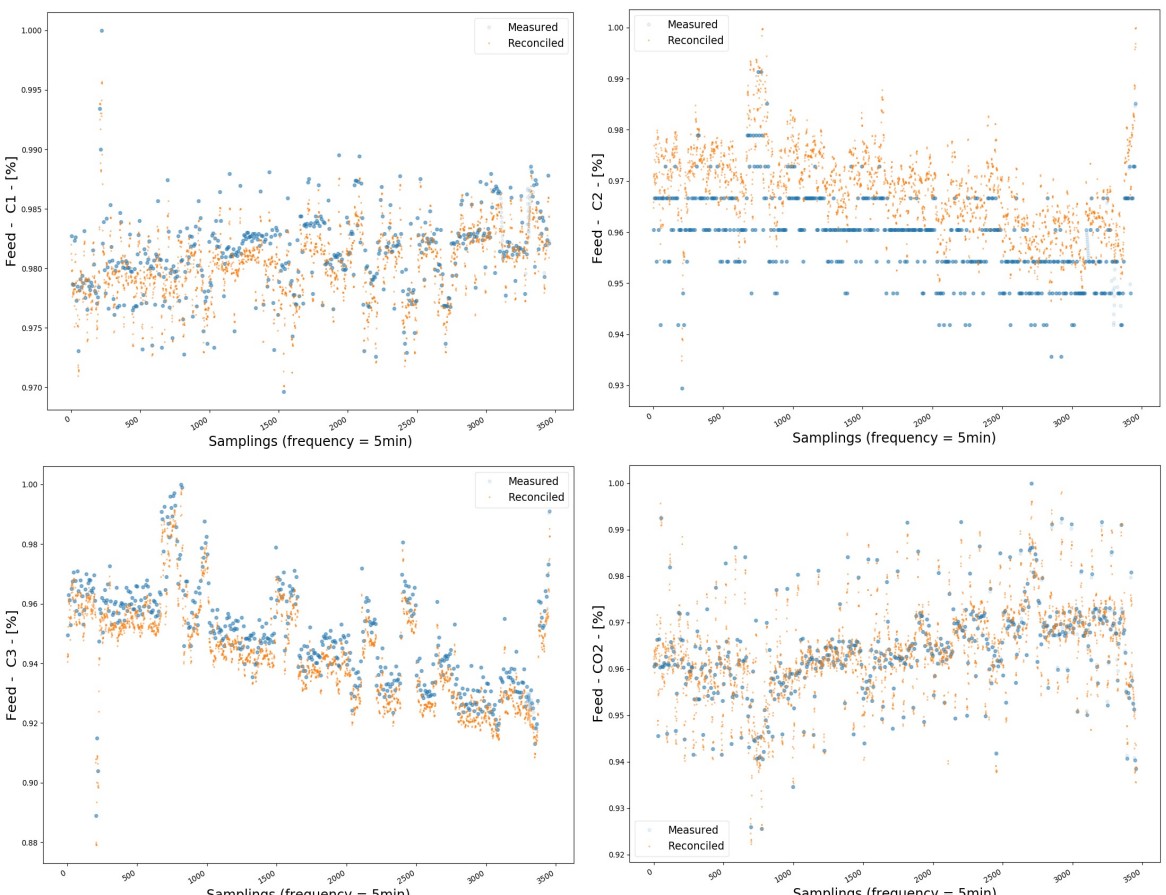

**Figure 12.** Offline data reconciliation for compositions $C_1$, $C_2$, $C_3$, and $CO_2$ in the feed stream.

Figure 13 illustrates the measured, reconciled, and estimated data for the flowrates. Another major advantage of DR is to identify the occurrence of systematic deviations in measurements, often caused by miscalibrated instruments. Figure 13 shows the occurrence of bias for the feed and residue flowrates.

Figure 14 shows the sum of residuals with respect to the mathematical model. The total squared residual is a measure of corrections that were needed in order to reconciled variables to satisfy the mass and energy balance equations.

Based on the previous results, it could be concluded that the DR procedure presented good performance and advantageous aspects for monitoring of the process. Monitoring processes with statistically treated information, detection of measurement bias, and identification of poor

instrumentation performances constitute good tools for diagnosing the states of the analyzed process and respective instrumentation.

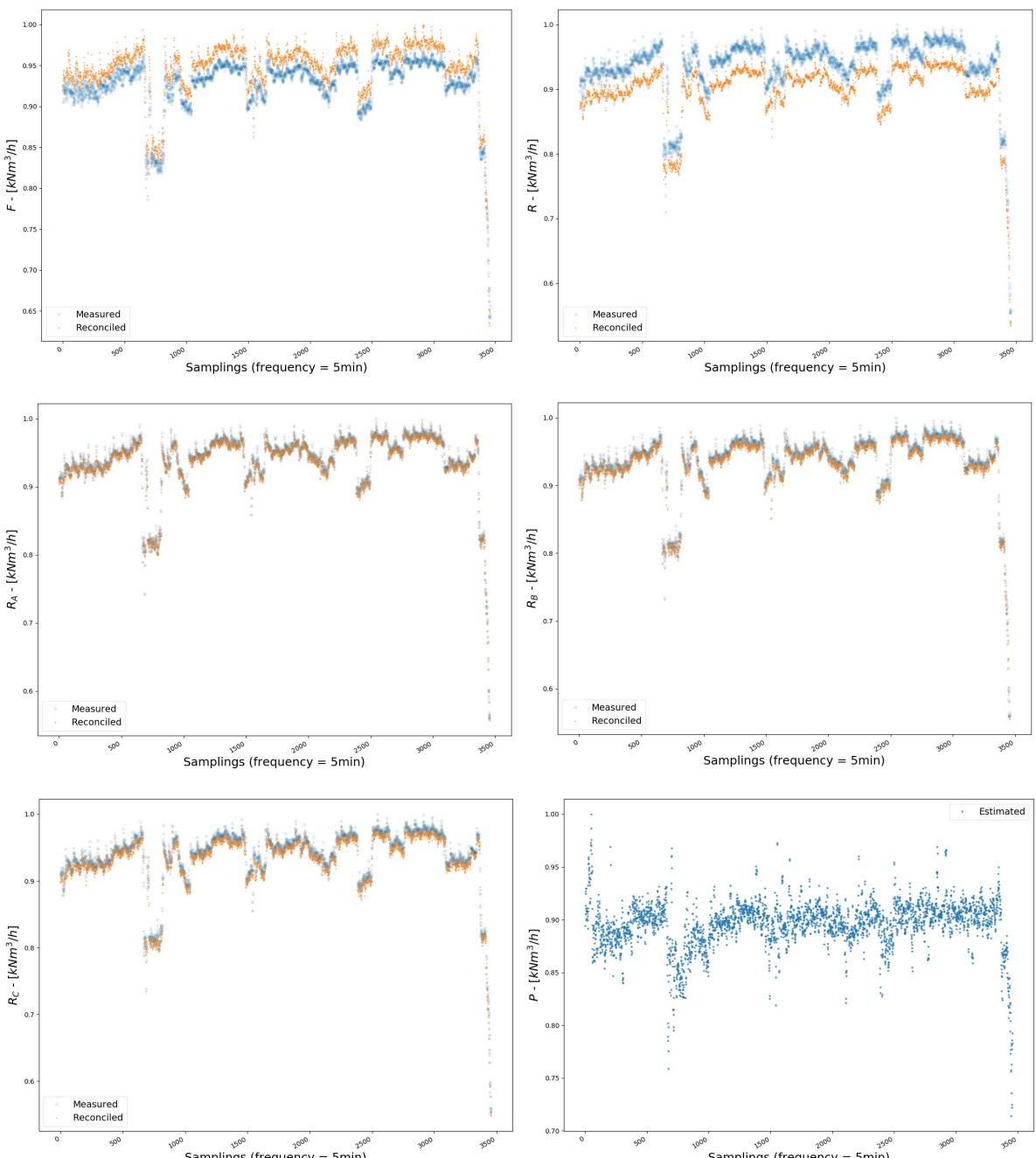

**Figure 13.** Offline data reconciliation of flowrates.

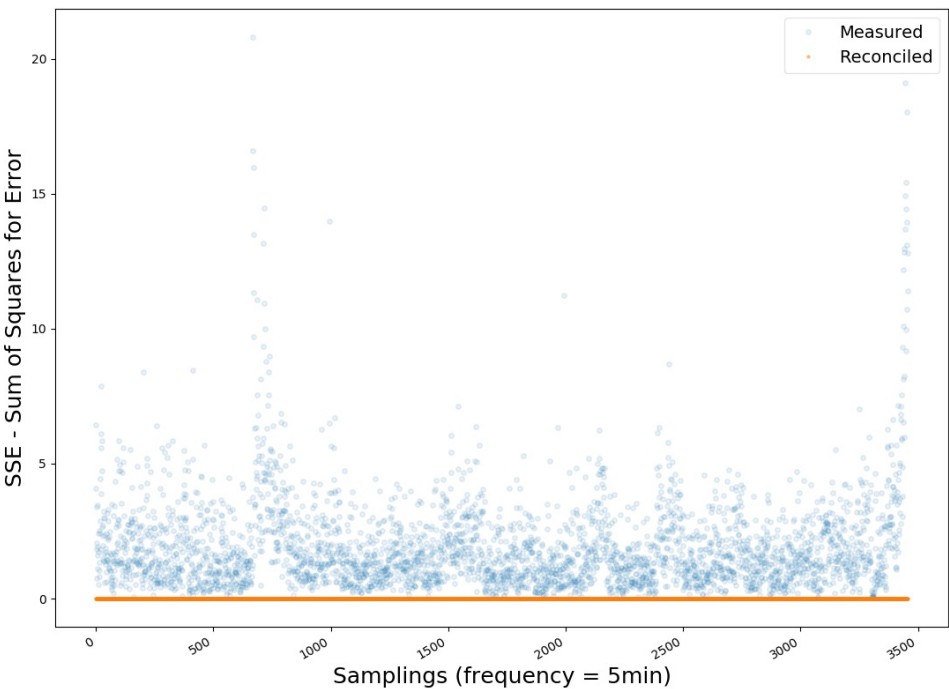

**Figure 14.** Sum of squares of model deviations (residuals) during offline data reconciliation.

### 3.3. Gross Error Detection

Procedures for removal of gross errors were implemented for online and real-time DR. These procedures were based on statistical tests using a moving variance window. Figures 15–20 illustrate a case of gross errors in which the procedure proved to be robust. However, it is possible to observe that the gross errors that affected the compositions of the retentate flowrate influenced the reconciliation of the feed flowrate. This occurred because of the well-known "smearing effect" when the DR procedure was performed with the WLS estimator (non-robust), even when variance adjustment was performed [48]. As gross error measurements were observed for a short period of time, it was not possible to detect the main source of the problem in the analyzed data set. Nevertheless, the occurrence of the problem was reported for maintenance teams for evaluation of measurement consistency.

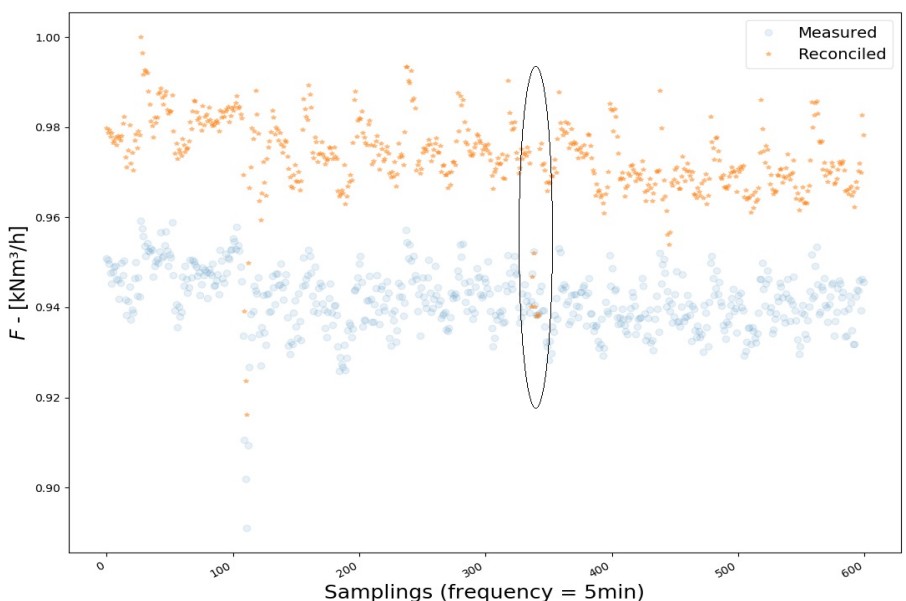

**Figure 15.** "Smearing" effect during the DR of feed flowrates.

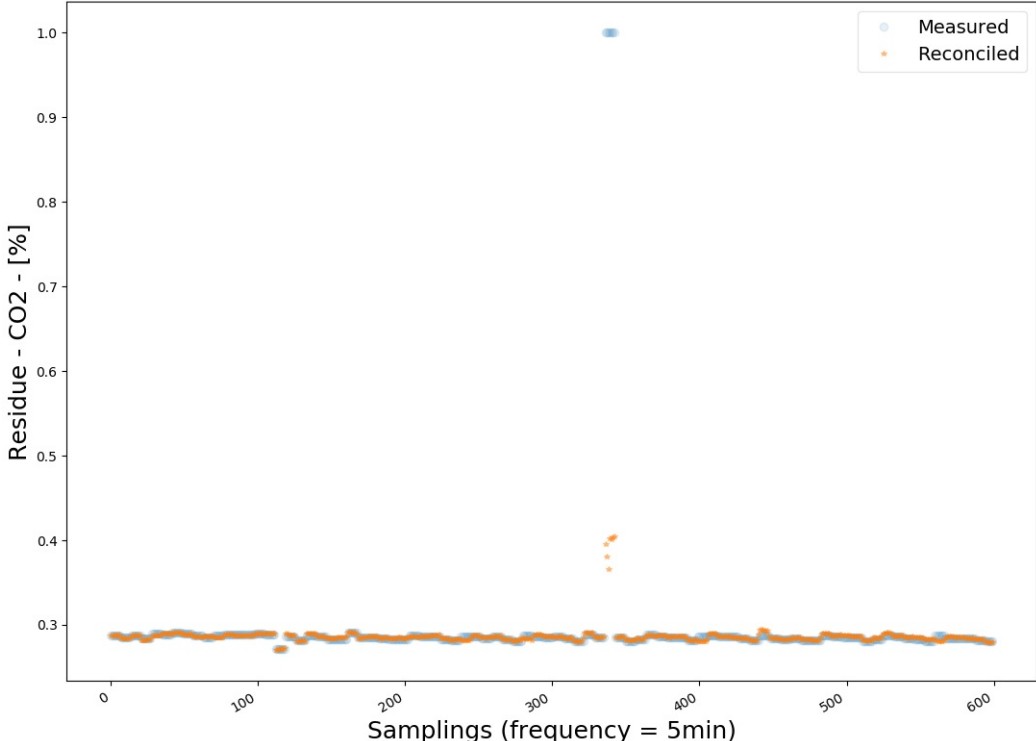

**Figure 16.** Gross Error Detection—$CO_2$ in the feed stream.

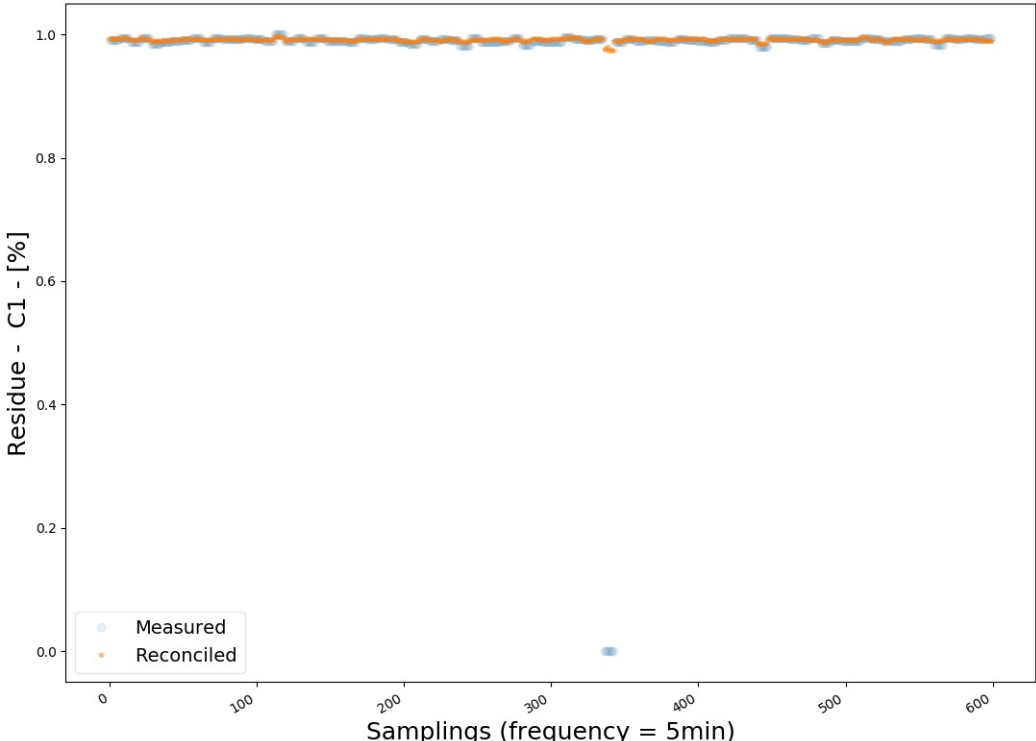

**Figure 17.** Gross Error Detection—$C_1$ in the feed stream.

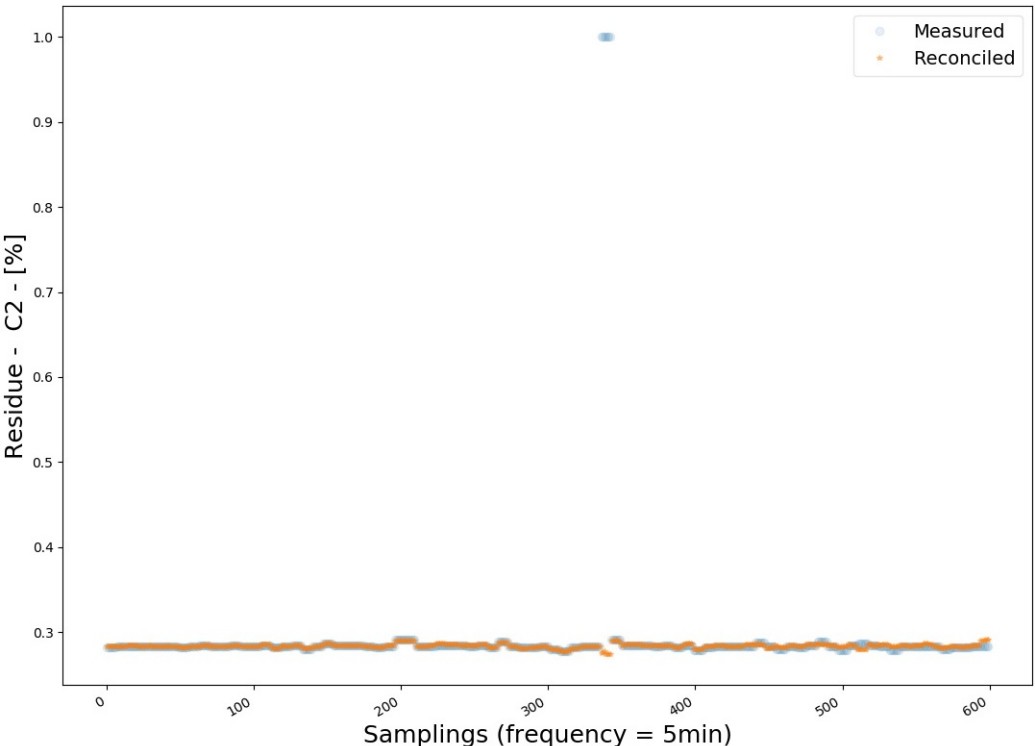

**Figure 18.** Gross Error Detection—$C_2$ in the feed stream.

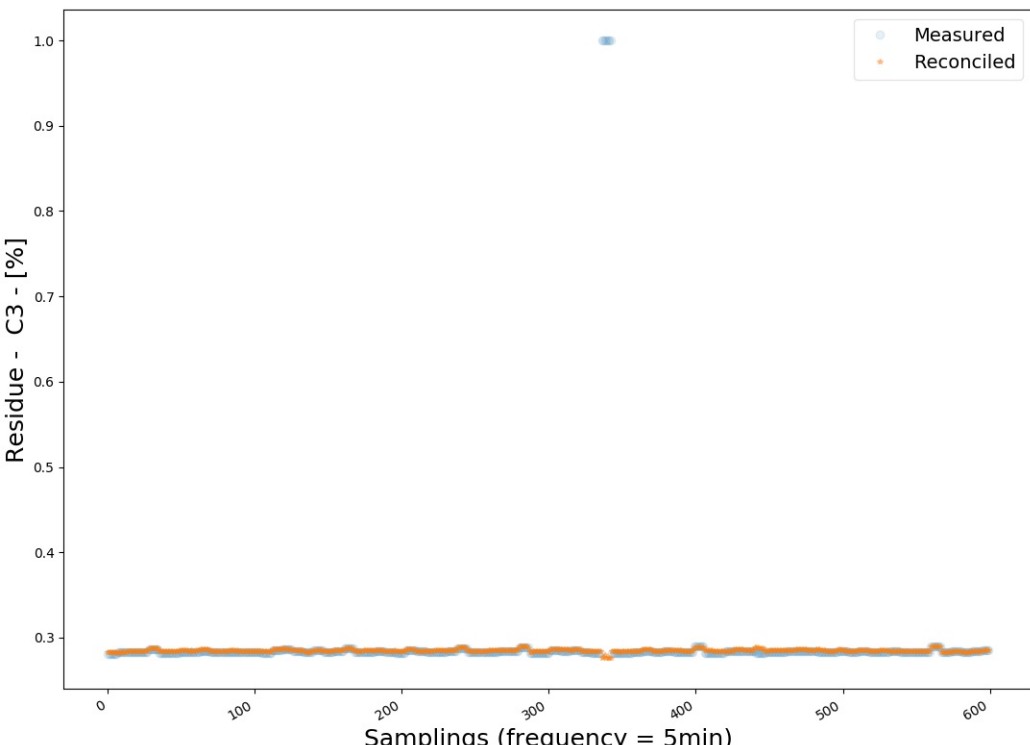

**Figure 19.** Gross Error Detection—$C_3$ in the feed stream.

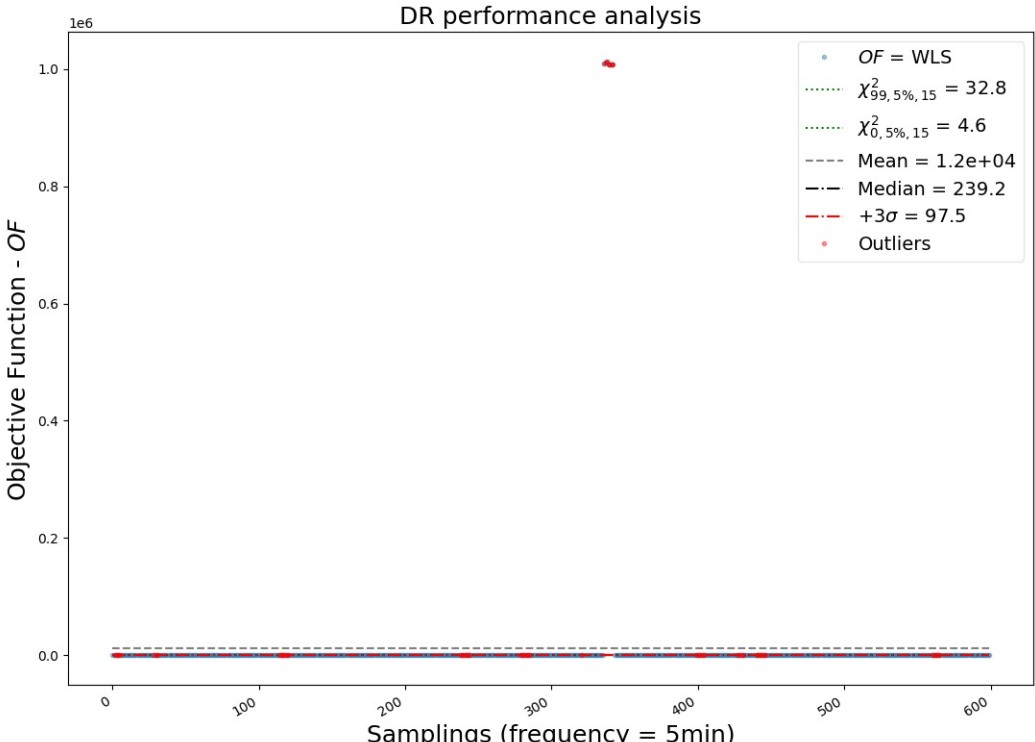

**Figure 20.** Gross Error Detection—DR performance analysis.

It is important to note that the statistical tests were implemented only for the compositions. The fact is that operational changes hindered the test because in many cases the test interpreted operational changes as outliers. Figure 21 illustrates that data reconciliation was effective and performed well after several operational changes.

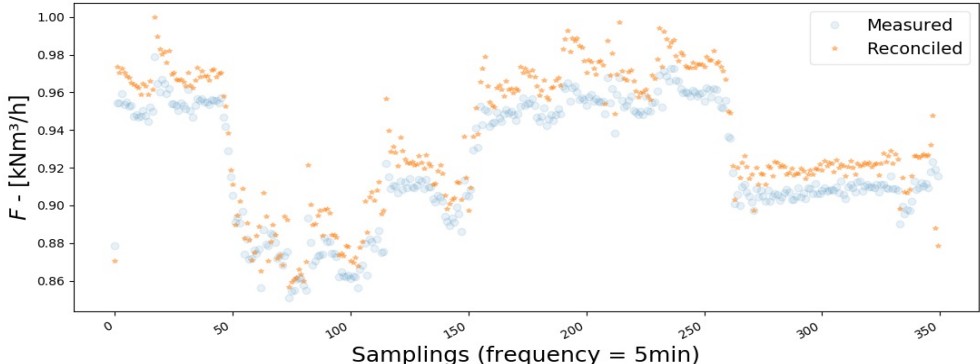

**Figure 21.** Monitoring through data reconciliation.

An important advantage of the moving variance window was to avoid the interruption of the online DR procedure due to measurement problems. These failures occur more frequently with compositions measured online through gas chromatography. These measurement failures cause missing data and, consequently, occurrence of series of constant values. As a result, the signal loses variability, preventing the realization of DR. Figures 22 and 23 illustrate cases of variable freezing caused by missing data.

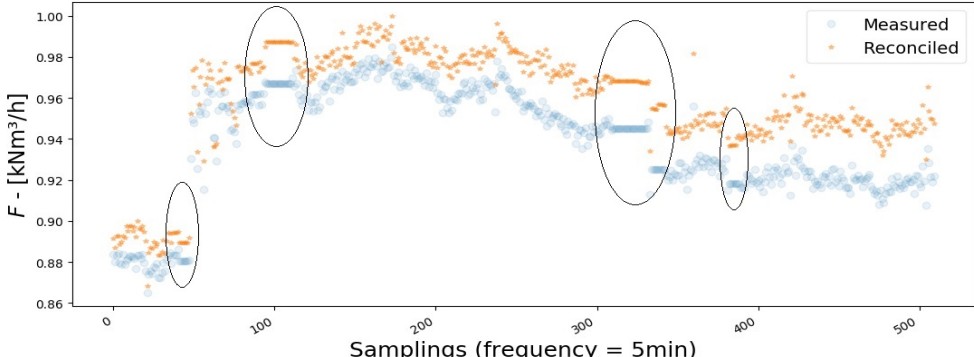

**Figure 22.** Measurement failures: missing data and frozen values of feed flowrate.

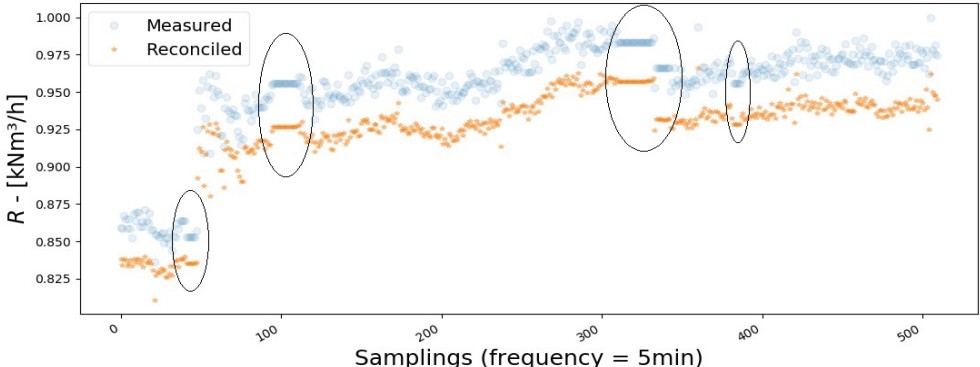

**Figure 23.** Measurement failures: missing data and frozen values of residue flowrate.

The analysis of bias can be performed through dynamic bar graph monitoring, illustrating the magnitude of the errors of each variable. Figure 24 informs the magnitudes of the systematic deviations from the median, that is, how many times the reconciled variable deviated from the measured median value. Systematic deviations that are larger than three times the value of the standard deviation can be regarded as a bias. Therefore, analyzing Figure 24, five variables with measurement biases could be observed: N2 (feed); N2 (residue); C8 (permeate); feed flowrate; and residue flowrate. Generally, biases can indicate the occurrence of unbalanced measurements, calibration problems, and instrument malfunctioning. For this reason, the obtained results were relayed to maintenance teams for evaluation of the instrumentation performances.

Figure 25 analyzes the performance of the DR, presenting the value of the OF, which represents the degree of correction of the reconciliation. The two green dotted lines represent the region where a normal distribution is expected for the errors of all measured variables. The region above the red dotted line indicates the samples that were subject to large corrections during the reconciliation step. Therefore, it is reasonable to consider the possible occurrence of outliers when the obtained value of OF deviated more than three standard deviations from the median value.

A test to observe the influences of biases on the analysis and performance of DR was performed. Figures 26 and 27 illustrate the same analyses performed for the same time window, as presented in Figures 24 and 25, but after identification and compensation of outliers. It can be observed that outliers significantly affected the average OF value. Based on Figures 24–27, it can be said that the analysis of bias and outliers performed very well. Figure 27 also illustrates the benefits of bias and outlier adjustments, as objective function values were reduced significantly and shed light on the existence of persistent outlier measurements. This reinforces the importance of bias and gross error diagnosing and the necessity to involve maintenance teams for evaluation of the instrumentation performances.

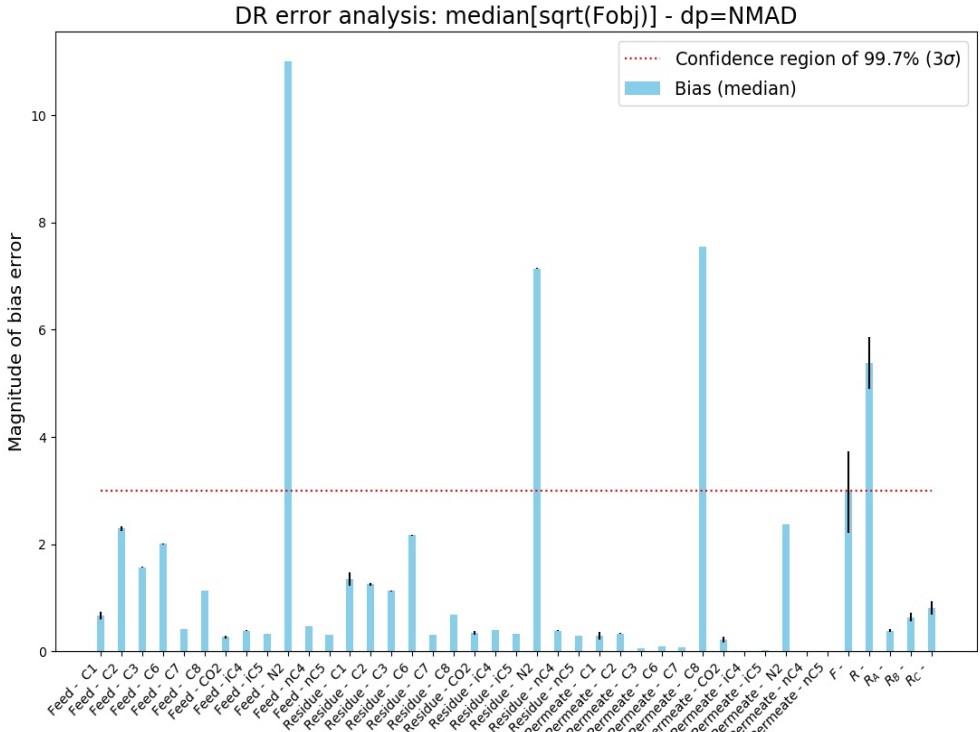

**Figure 24.** DR analysis without bias compensation.

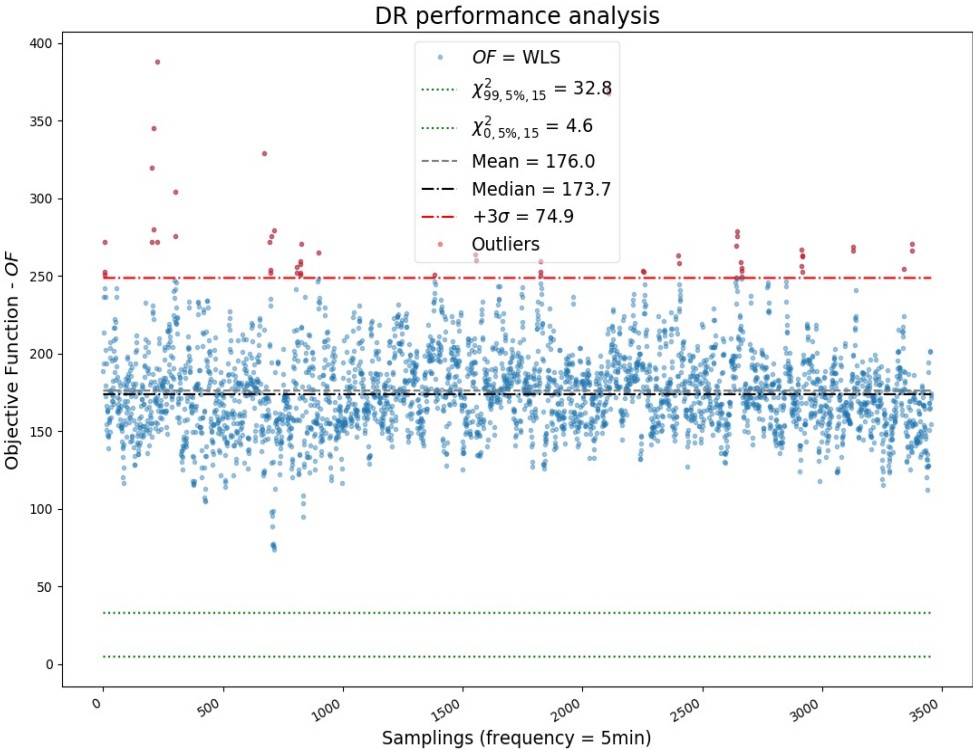

**Figure 25.** DR performance analysis without bias compensation.

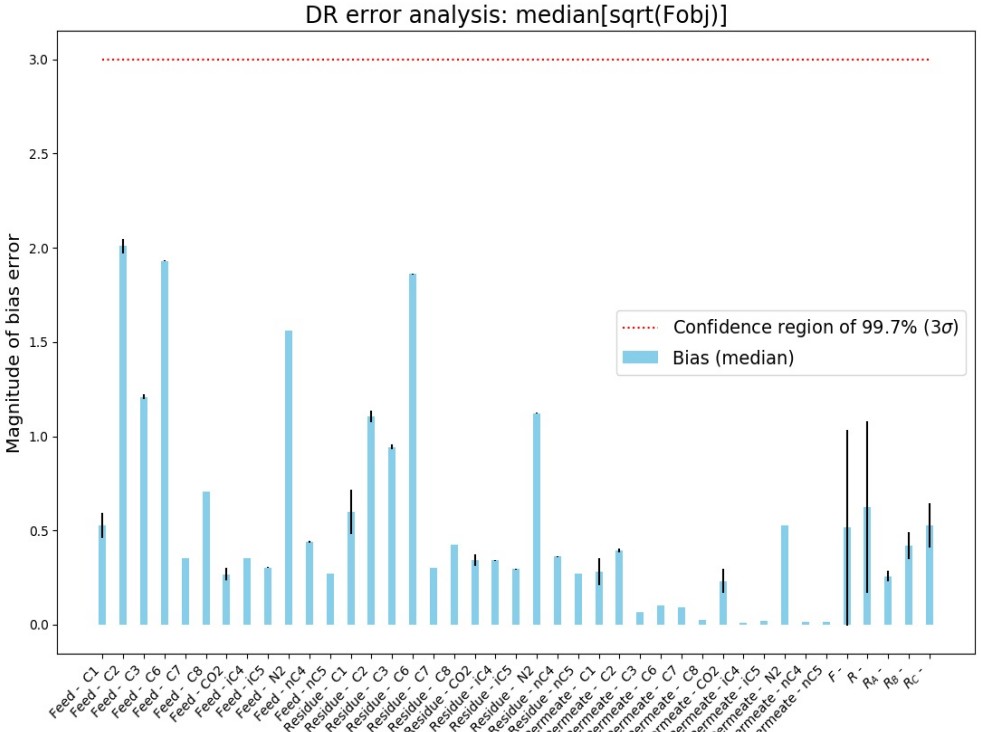

**Figure 26.** DR analysis with bias compensation.

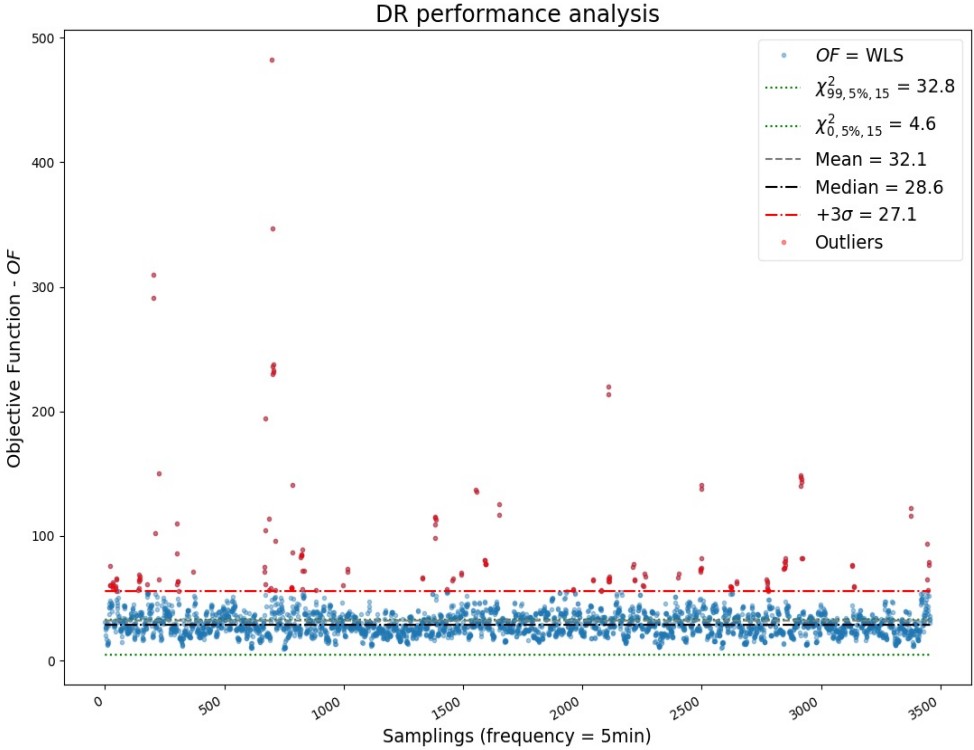

**Figure 27.** DR performance analysis with bias compensation.

### 3.4. Monitoring

In each new cycle of data acquisition, the code runs in sequence the pre-treatment, statistical tests and outlier compensation, data reconciliation with the permeate flowrate estimation, and finally the energy balance to calculate the temperature of the permeate flowrate. The first inferred variable was the permeate flowrate, estimated within the data reconciliation procedure. Figure 28 illustrates the real-time monitoring of the inferred variable.

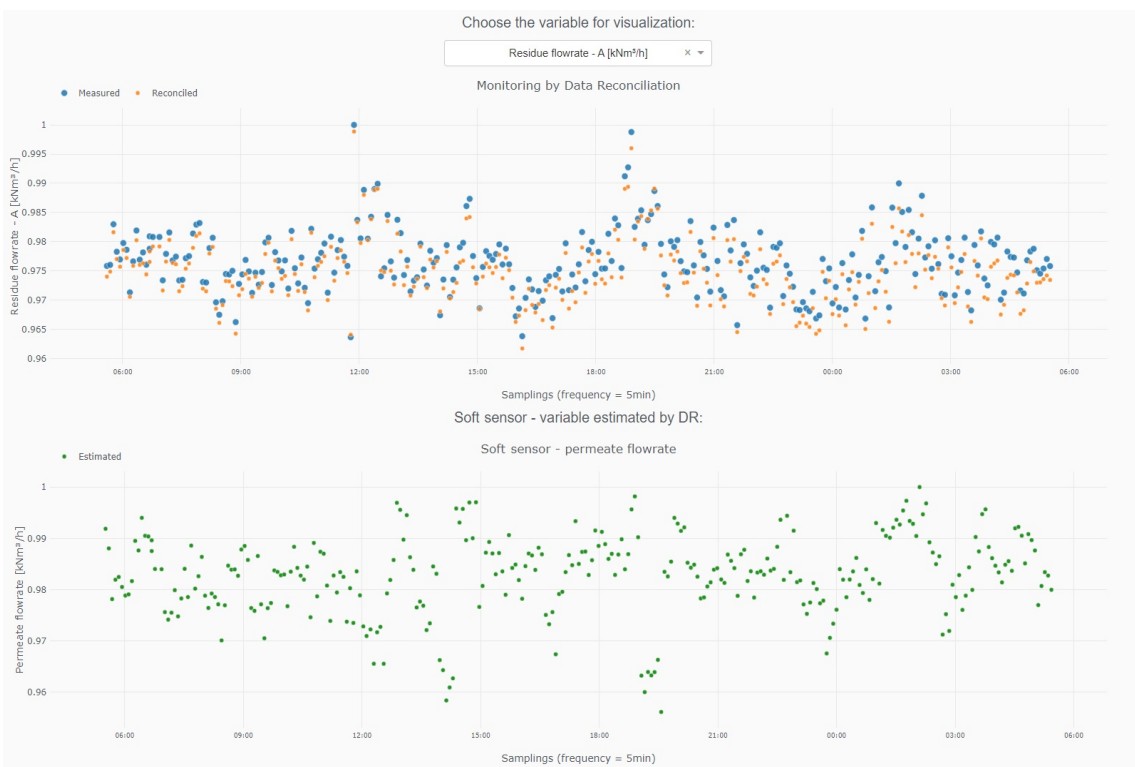

**Figure 28.** Real-time soft sensor—permeate flowrate.

The second inferred variable was the temperature of the permeate. At this stage, it was not possible to reconcile data due to the lack of redundancy of measured variables. Thus, this variable was inferred without the proper statistical treatment by the DR stage. Figure 29 illustrates part of the web application (web-app) where the user interacts with the interface. The variables can be selected through a dropdown menu. In addition, the application provides graphs of gross error analysis (Figure 30), visualization of inferred variables, and a button to start and stop monitoring. Figure 31 illustrates the three temperatures of each stream, during the testing period of the web-app. The permeate temperature was calculated with help of the energy balance and the regions without data are the days when the web-app was paused. All monitoring data are saved and can be read and analyzed offline, as in the case of Figure 31.

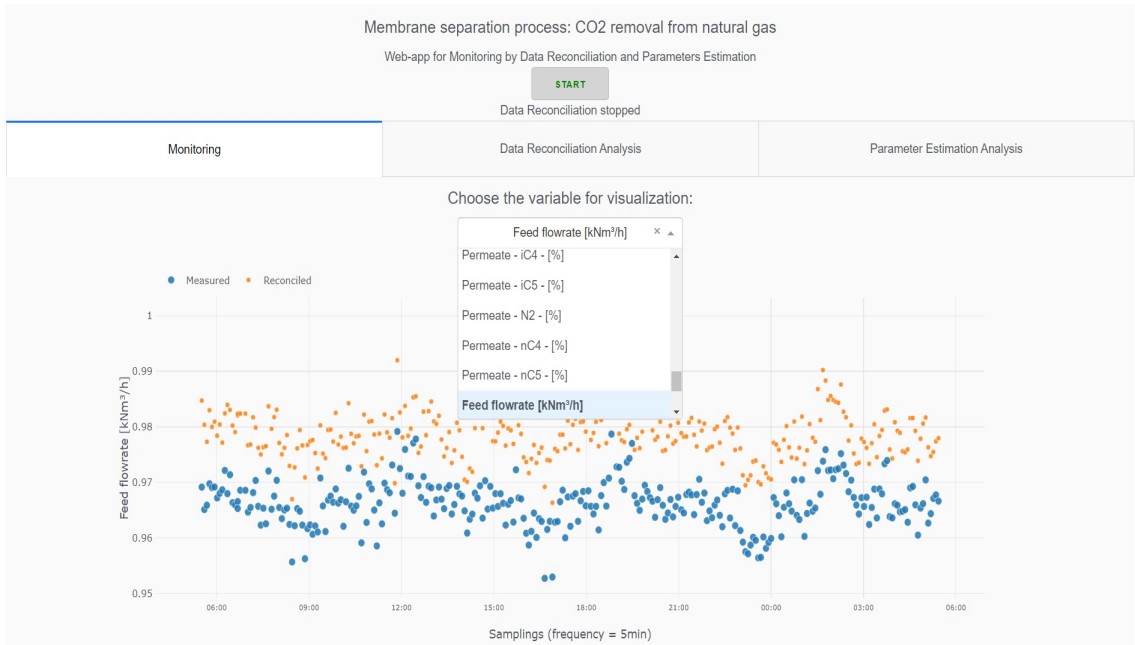

**Figure 29.** Part of the web-app: variables measured and reconciled.

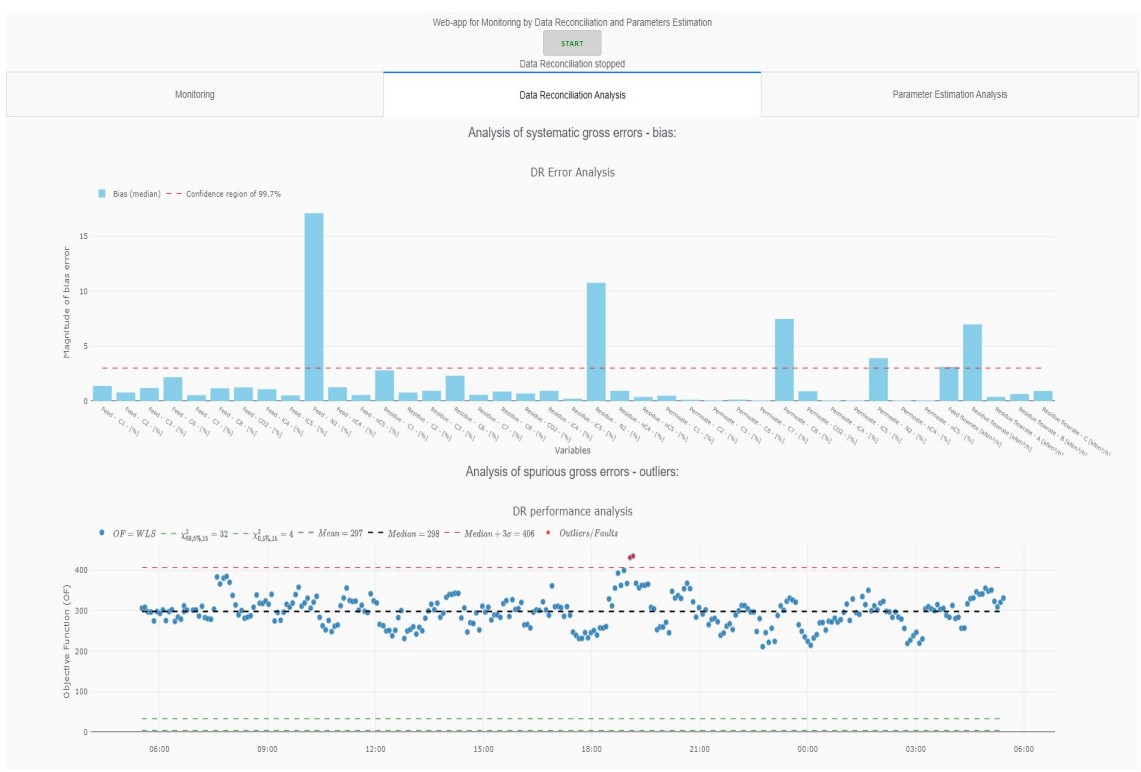

**Figure 30.** Part of the web-app: data reconciliation analysis.

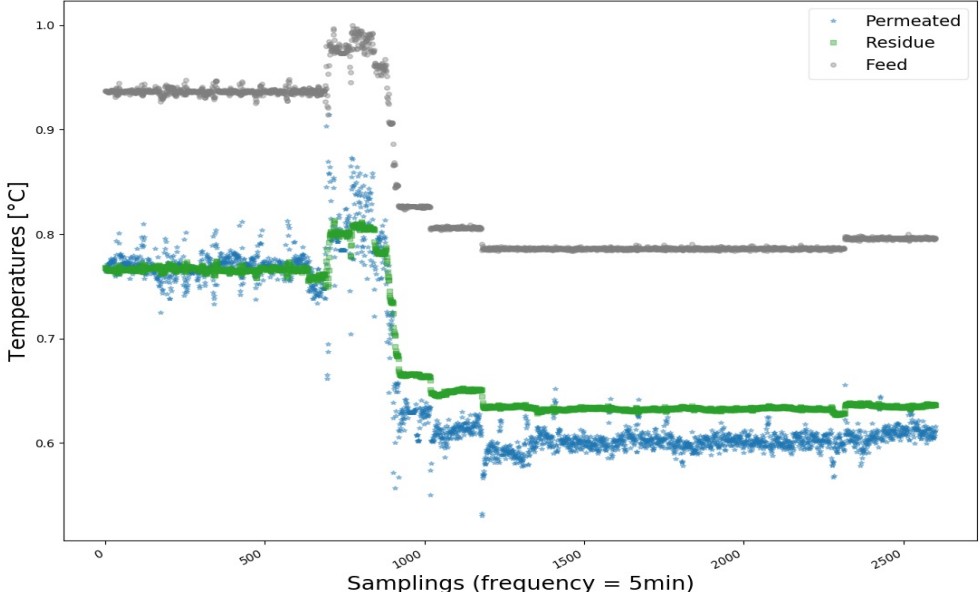

**Figure 31.** Offline data analysis: Temperatures in the testing period.

Therefore, the temperature inferred by the calculation of the energy balance showed good accuracy in relation to reported offline measurements, which demonstrates the importance of DR for the treatment of the variables used in the soft sensor.

## 4. Conclusions

A methodology was developed and implemented for the first time in the form of a web application to allow the monitoring of membrane separation processes online and in real time, making use of statistical techniques for treatment of process data. The proposed methodology comprises the following stages: (i) pre-treatment and characterization of process data; (ii) data reconciliation of process data to minimize measurement uncertainties, with the aid of mass balance equations; (iii) detection of systematic deviations for identification of process malfunctions; and (iv) observation of unmeasured variables (working as a soft sensor or digital twin). The pre-treatment and data characterization steps were fundamental for the understanding and correct formulation of the problem. The characterization step can find wide application, as this procedure can be applied in any chemical process. This step is essential for the appropriate selection of data reconciliation techniques and gross error detection procedures. After that, the proposed data reconciliation and gross error detection steps showed robustness, good performance, and speed. The proposed scheme was based on detailed steady-state balance equations, validated after proper characterization of actual operation data. The numerical procedures were validated offline and then implemented online and in real time for the first time, allowing the successful identification of measurement biases and outliers and providing estimates for unmeasured data. The developed procedures can be used for online and real-time detection of process faults and process diagnosing. In addition, the procedure provides reliable data for future stages of simulations and parameter estimation, allowing the implementation of digital twins, as the model proposed in part I of this research project. Production Management System and Enterprise Resource Planning steps can also benefit from availability of more reliable data, and variables inferred by a soft sensor. Therefore, the main advantages of the procedure are reliable data handling, diagnosis of gross errors/failures, and real-time monitoring of the process.

**Author Contributions:** D.Q.F.d.M.: Conceptualization, Methodology, Software, Validation, Data curation, Writing—original draft, Writing—review and editing; M.C.C.d.S.: Software, Validation, Formal analysis; T.B.F.: Software, Validation, Formal analysis; T.K.A.: Resources and Funding acquisition; F.C.D.: Resources and Funding acquisition; P.H.T.: Resources and Funding acquisition; J.C.P.: Supervision, Project administration, Writing—review and editing. All authors have read and agreed to the published version of the manuscript.

**Funding:** This work was supported by the Petróleo Brasileiro SA (Petrobras); Conselho Nacional de Desenvolvimento Científico e Tecnológico (CNPq) and Coordenação de Aperfeiçoamento de Pessoal de Nível Superior (CAPES).

**Acknowledgments:** The authors thank Petrobras, CNPq, and CAPES for the financial support to this work, as well as for covering the costs to publish in open access.

**Conflicts of Interest:** The authors declare that they have no known competing financial interests or personal relationships that could have appeared to influence the work reported in this paper.

## Abbreviations

The following abbreviations are used in this manuscript:

| | |
|---|---|
| GED | Gross error detection |
| DR | Data reconciliation |
| WLS | Weighted least squares |
| PI | Plant information |
| HDF5 | Hierarchy Data Format version 5 |
| NaN | Not a number |
| LL | Lower limit |
| UL | Upper limit |
| OF | Objective function |
| GT | Global test |
| NMAD | Normalized median absolute deviation |
| ACF | Autocorrelation functions |
| PACF | Partial autocorrelation functions |
| CCF | Cross-Correlations Function |

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
