# Peer review of "Modeling of Spiral Wound Membranes for Gas Separations—Part II: Data Reconciliation for Online Monitoring"

_processes, doi:10.3390/pr8091035_

Round 1
Reviewer 1 Report
Overall a very interesting and a well written manuscript focusing on modelling for monitoring membrane separation processes on a real time basis. In my opinion, the manuscript is structured very well, has provided in detail and clear explanations of the methodology, and offers a critical literature review and discussion of its findings. Nonetheless, I believe the work could still benefit from the following minor revisions:
- Please re-write the abstract, as I believe it should better recapitulate the key contributions of the manuscript. In my first read, the abstract did not provide a concise summary of all aspects of the research undertaken.
- The paper is generally written very well, yet I think it could still benefit from another round of proof-reading; to give an instance, in Line 37 ‘in line’ should be have been written as ‘Online’,
- Figure 28 which seems to be part of the centralised web-app does not illustrate the drop-down menus which is mentioned within an earlier paragraph; please clarify this either within the text or replace the figure,
- The methodology could be part of a plant centralised digital twin (although you have touched on this in a few instances throughout the paper); I would suggest that you further strengthen the conclusion section by briefly discussing how the real-time/semi-real-time nature of the proposed methodology could inform a real-world digital twin as part of a further work.
Thank you.
Author Response
We thank Reviewer#1 for his/her suggestions. Reviewer#1’s suggestions have been accepted and modifications have been introduced into this revised version of the manuscript as proposed.
Author's notes to reviewer in annex.

Reviewer 2 Report
The article describes a methodology based on data reconciliation to monitor membrane separation processes reliably, on line and in real time. The work is a continuation of an article published in 2020 in the Journal of Membrane Science entitled "Modeling of spiral wound membranes for gas separations. Part I: An iterative 2D permeation model". After reading the text, I pay attention to the following flaws:
- Captions to all figures are "Figura". It should be "Figure".
- Both axes are not captioned in Figures: 7, 8, 10, 11, 12, 13, 14, 15, 16, 17, 18, 19, 20, 21, 22, 27, 28, and 29. For example, the y axis in Figure 7 is not captioned. The situation is similar with the other Figures.
- In Figure 23, some bias values are far beyond the confidence line. What is the interpretation of such a phenomenon?
- In Figure 25 all bias values do not exceed the confidence line. How do the authors interpret this phenomenon?
- In Figures 16, 17, 18 and 19 the Residue [%] is much larger in the area for a value of around 350. How to interpret this phenomenon?
- At the end of the conclusion section, the authors write “The developed procedures can be used for on-line and real-time detection of 472 process faults and process diagnosing”.
- "On-line" and "real-time" information seems to be important, because it is related to the speed of operation of the diagnostic system. The speed of operation of a complex cyber physical system is influenced not only by the algorithm, but also by hardware factors and data transfer. Have the authors considered these aspects in their research? What is the importance of the speed of operation of the diagnostic system in relation to the research object?
- In the Introduction section, the authors define the purpose of the research - “The main objective of the present work is to present a methodology for monitoring of membrane separation processes on line and in real time, making use of statistical techniques for treatment of process data.” The mere presentation of the methodology does not sound convincing. Usually, the objective of research is defined in the context of meeting a need, solving a research problem. Therefore, it is worth correcting the description of the research purpose. Apart from the goal, I would also expect a few sentences about what new research brings to the present state of knowledge. Novelties are important. Especially if they are presented against the background of other, alternative solutions.
- The Conclusions section is too laconic, especially since the Results section is very extensive and contains many Figures. In this section the authors should discuss the results and how they can be interpreted in perspective of previous studies and of the working hypotheses. The findings and their implications should be discussed in the broadest context possible. Future research directions may also be highlighted. Meanwhile, many sentences in this section are shallow. There is no highlight of the advantages of the new approach against the background of known and widely used methods. No information if the main goal of the research was achieved. There are no convincing arguments and comparisons related to the results obtained and testifying in favor of the new method. A better title for this section would be ‘'Discussion and conclusions'.
Author Response
We thank Reviewer#2 for his/her suggestions. Reviewer#2’s suggestions have been accepted and modifications have been introduced into this revised version of the manuscript as proposed.
Author's notes to reviewer in annex.

Reviewer 3 Report
The authors present their work on data reconciliation work with modeling of spiral wound membranes for gas separations. The work is of relevance to the researchers in the field. I have the following minor comments:
- The authors should thoroughly check the manuscript for spelling errors and grammatical accuracy. For example on page 3, line 79: 'desviations' should be corrected to 'deviations'
- The authors could consider providing comments on their future work in the concluding paragraph
Author Response
We thank Reviewer#3 for his/her suggestions. Reviewer#3’s suggestions have been accepted and modifications have been introduced into this revised version of the manuscript as proposed.
Author's notes to reviewer in annex.

Round 2
Reviewer 2 Report
After revision, the article is ready for publication in its present form.